# Stability of person-specific blood-based infrared molecular fingerprints opens up prospects for health monitoring

Marinus Huber [1,2✉], Kosmas V. Kepesidis[1], Liudmila Voronina [1,2], Maša Božić[1], Michael Trubetskov [2], Nadia Harbeck[3], Ferenc Krausz[1,2,4] & Mihaela Žigman [1,2,4✉]

Health state transitions are reflected in characteristic changes in the molecular composition of biofluids. Detecting these changes in parallel, across a broad spectrum of molecular species, could contribute to the detection of abnormal physiologies. Fingerprinting of biofluids by infrared vibrational spectroscopy offers that capacity. Whether its potential for health monitoring can indeed be exploited critically depends on how stable infrared molecular fingerprints (IMFs) of individuals prove to be over time. Here we report a proof-of-concept study that addresses this question. Using Fourier-transform infrared spectroscopy, we have fingerprinted blood serum and plasma samples from 31 healthy, non-symptomatic individuals, who were sampled up to 13 times over a period of 7 weeks and again after 6 months. The measurements were performed directly on liquid serum and plasma samples, yielding a time- and cost-effective workflow and a high degree of reproducibility. The resulting IMFs were found to be highly stable over clinically relevant time scales. Single measurements yielded a multiplicity of person-specific spectral markers, allowing individual molecular phenotypes to be detected and followed over time. This previously unknown temporal stability of individual biochemical fingerprints forms the basis for future applications of blood-based infrared spectral fingerprinting as a multiomics-based mode of health monitoring.

[1] Department of Physics, Ludwig Maximilian University of Munich, Garching, Germany. [2] Max Planck Institute of Quantum Optics, Garching, Germany. [3] Department of Obstetrics and Gynecology, Breast Center and Comprehensive Cancer Center (CCLMU), Hospital of the Ludwig Maximilian University (LMU), Munich, Germany. [4] Center for Molecular Fingerprinting (CMF), Budapest, Hungary. ✉email: huber.marinus@physik.uni-muenchen.de; mihaela.zigman@mpq.mpg.de

Probing of systemic human biofluids such as blood serum and plasma offers a potential means of monitoring the health status of individuals[1,2]. Molecular fingerprinting of blood-based biopsies via infrared vibrational spectroscopy[3–6] constitutes one possible way of realizing this potential. However, whether or not infrared molecular fingerprints (IMFs) are sufficiently stable over time to allow for health monitoring has not yet been assessed, nor have standard ranges for IMFs of healthy populations been determined. Human blood composition is influenced not only by a multitude of physiological states, but also by genotypic variation, lifestyle, age, environmental factors, nutritional status, drug consumption, and even metabolites produced by the symbiotic microflora[7–10]. Hence, any liquid-biopsy-based approach to health state monitoring must take natural biological variability, and the reference ranges for parameters that are sensitive to the physiological state of the organism, into account[2,8–10]. These parameters can be either individual analytes or specific features in a spectral fingerprint. The aim of this study is to evaluate the stability of IMFs and their spectral markers over time and provide a general understanding of the range of blood-based biological variability across molecular species, which is a vital prerequisite for any future application of molecular fingerprinting in health monitoring or disease detection.

Analytical "omics" approaches for molecular profiling, such as mass spectrometry (MS), nuclear magnetic resonance (NMR) spectroscopy, or DNA/RNA-sequencing methods, have led to the discovery of numerous blood-based biomarkers as candidates for disease detection and treatment monitoring[1,11–18]. Although sensitive and specific, most of these techniques focus on a single molecular group in a given context: i.e., they measure either proteins[12], or small molecule metabolites[13], or lipids[14], or DNA[15], or RNA[16]. However, probing of different molecular classes in parallel ("multiomics") may better capture patterns of characteristic molecular changes and thus allow one to define significant pathophysiological transitions[1,17,18]. Infrared vibrational spectroscopy[3–6] probes vibrations of the structural backbones of all molecular species in a sample. The frequencies of those vibrations depend on the atomic composition, structure, and strength of the chemical bonds in the molecules. Thus, infrared spectroscopy has the inherent advantage of being sensitive to all functional groups in organic samples[5,6]. Unfortunately, spectral overlap of molecular responses and limited sensitivity of commercially available infrared spectrometers allows vibrational spectroscopy to quantify only the most abundant substances of highly complex bioliquids so far[19,20]. However, new spectroscopic schemes allow to overcome current limitations in sensitivity and have the potential to significantly increase the range of detectable molecular concentrations[21,22].

When applied to liquid biopsies, vibrational spectroscopy provides an IMF, which is potentially specific for a molecular blood phenotype and can therefore serve as a marker for an individual's state of health. Fourier-transform infrared (FTIR) spectroscopy has demonstrated the potential of spectral fingerprints for disease diagnostics (e.g., Alzheimer's disease[23–25], prostate[26], lung[27], breast[28], liver[29], and brain cancers[30]) as well as for tracing the evolution of metabolic changes under exercise-induced conditions in athletes (sports medicine)[31–33]. FTIR spectroscopy of biofluids also has been used for disease monitoring in animal models[34,35] and blood biopsies from patients[36,37]. However, to the best of our knowledge, no attempts have yet been made to assess the stability of IMFs of a healthy, non-symptomatic human population over time. Thus, the inevitable biological variability of human biopsies relevant to any health monitoring approach remains unexplored.

This study addresses questions that are fundamental for the applicability of infrared fingerprinting in health monitoring: First, we test whether infrared spectral fingerprints can be reproducibly and directly obtained from bulk liquid blood serum and plasma samples, and we determine the range of natural biological variation of IMFs from individual volunteers over time (within-person variation). Second, we quantitatively relate the variation of the IMFs over time for any given individual to the degree of variability between different individuals (between-person variation) and to operational variabilities inherent to clinical practice. We address these questions in a prototypical human clinical study cohort, quantify the analytical measurement error, and relate this to the variation between four different clinical centers (inter-clinical variability). Our study provides evidence for the existence of detectable person-specific IMFs of liquid-phase human blood samples. This lays the foundations for IMF as a promising discriminative and non-invasive method for health monitoring in the future.

## Results

To assess whether infrared vibrational spectra obtained from human blood in the liquid phase have properties that permit its use for health monitoring, we systematically quantify the within-person and the between-person variabilities of IMFs and relate these to the analytical and the clinical error. To this end, we analyzed prospectively collected samples of blood serum and blood plasma from 31 healthy, non-symptomatic human individuals. A detailed breakdown of the study participants is given in Supplementary Table 1. Blood was drawn from each individual in the cohort on 13 different days over a period of 7 weeks (once every 3–4 days), and once or twice after 6 months (Fig. 1a and Supplementary Table 2). In addition, the influence of measurement variability as well as blood collection and sample-handling processes were characterized. This combined error was evaluated in a separate study by comparing blood samples obtained at four different clinics from five individuals within <4 h. Well-defined standardized protocols for blood drawing, sample processing, and sample storage, applicable to routine medical practice were used throughout the study. This allowed us to evaluate variability caused by variations in blood drawing and sample processing, as well as short-term changes in blood composition[38]. The chosen study design represents a typical prospective longitudinal clinical study setting for health monitoring.

**Infrared molecular fingerprints of liquid blood plasma and serum.** We measured the infrared absorption spectra of blood serum and plasma samples with an automated FTIR device. Serum and plasma were transilluminated as native liquids in a thin flow-through cuvette (~8 μm path length) to mitigate the effects of strong absorption by water (see also Supplementary Figure 1). A measurement of a single sample took <5 min. In comparison with measurements of dried serum/plasma, this approach avoids major sample preparation steps and artefacts (e.g., the coffee-ring effect[4,39]) and preserves the native secondary protein structure, altogether increasing the reproducibility of the measurements as previously shown[40]. With this approach, we can record IMFs in a time- and cost-effective manner, with minimal sample preparation. The IMFs obtained covered the spectral range between 950 and 3000 $cm^{-1}$, which includes absorption bands characteristic for proteins (amide I/II, predominantly at 1548 $cm^{-1}$ and 1654 $cm^{-1}$), carbohydrates (mainly between 1000 and 1200 $cm^{-1}$), and lipids (1741 $cm^{-1}$, 2854 $cm^{-1}$, and 2929 $cm^{-1}$) (Fig. 1b).

There was an overall resemblance between the infrared spectra obtained from all study participants (Fig. 1b). The IMFs of blood plasma and serum are similar in overall shape, featuring the same characteristic absorption bands. This is not surprising, since

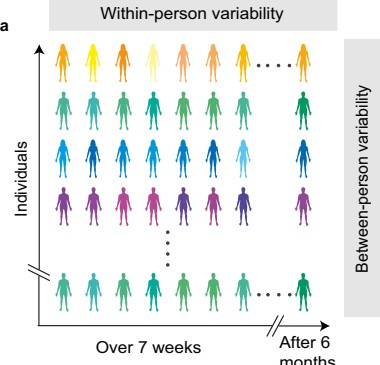

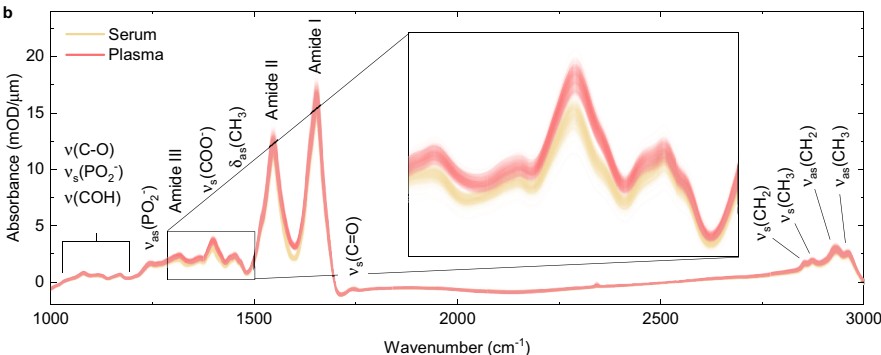

**Fig. 1 Study setup and overview. a** Experimental setup used for profiling of FTIR blood serum and plasma drawn from 31 healthy, non-symptomatic volunteers at up to 15 consecutive time points over the course of >6 months (see also Supplementary Table 2 for detailed information about the sampling time points). Same individuals are indicated as different shades of the same colors. **b** Unprocessed infrared absorption spectra of liquid blood sera (yellow) and plasma samples (red) measured from all individuals enrolled in the study. Inset: Close-up showing 636 individual traces of 318 measurements of blood sera and plasma each. Absorption peaks are associated with major molecular vibrations: $\nu$ stretching, $\delta$ bending, $s$ symmetric, and $as$ asymmetric vibrations.

plasma and serum share the vast majority of their molecular components. We find that the major difference between serum and plasma spectra is attributable to the ethylenediaminetetraacetic acid, which is added during plasma preparation and whose spectral features were readily recognizable (Supplementary Figure 2).

**Spectrally resolved variability of IMFs**. First, we determined the impact of natural biological variation on the acquired IMFs. To assess this variation quantitatively, we evaluated the magnitude of the time-dependent change (day-to-day and month-to-month) in the IMF of every single individuum in our study cohort (within-person variability) and the spread among individuals within the same population (between-person variability). Second, we compared them with variations arising from (minor) differences in blood collection and sample-handling processes (inter-clinical error) and to the evaluated error of the spectroscopic measurement.

Unprocessed infrared absorption spectra and their standard deviation owing to the variabilities caused by the above-mentioned effects show a similar dependence on wavenumber (Fig. 2a). This suggests that the variation of IMFs is dominated by differences in the total amount of molecules in the samples (e.g., owing to disparities in details of collection, handling, and processing) rather than by changes in their relative molecular composition. These uncertainties can be substantially reduced by additional spectral pre-processing, in particular by normalization of the measured IR spectra[41] (see Methods for details). When applied, this step reduces the relative inter-clinical variability and the relative measurement error to <1% and 0.1%, respectively, in most spectral ranges (Fig. 2b). Overall, the reproducibility of the measurements achieved here is better than what was previously shown with liquid or dried serum or plasma[40,42]. We, therefore, use pre-processed spectra for all further analyses.

Removing variations in overall biomolecular content brings to light the fact that within-person variation of molecular composition is much smaller than its spread across the cohort of healthy, non-symptomatic individuals, and that the inter-clinical variation is significantly lower than any of these biological variabilities in most of the spectral regions. We found only a few spectral regions in which the inter-clinical variability and within-person variability are comparable (Fig. 2b), and should thus be considered carefully when included further for analysis. Although the inter-clinical variability may be further reduced by improved protocols for sample collection, the spectral variability is already on a level,

which allows characterization of the change of a person's molecular IR fingerprint over time.

**Comparison of biological variability in blood serum and plasma**. To evaluate whether blood serum or blood plasma might be better suited for IMF-based clinical diagnostics, we compared the magnitude of biological variability in the two bioliquids after pre-processing their measured infrared absorption spectra. Spectrally resolved variability was averaged over the whole spectrum and the within-person variability for each person was evaluated individually. Although levels of between-person and inter-clinical variability were approximately the same for serum and plasma, the within-person variability of plasma was, on average, 24% higher than for serum samples (with a statistical significance of $p = 2.7 \times 10^{-4}$) (Fig. 2c). This shows that IMFs of plasma samples captures more of the variations in molecular composition over time than IMFs of serum samples do. Depending on whether this additional biological information is desired for the envisioned monitoring or diagnostic application, the use of one or the other medium may be preferable.

**Within-person and between-person variability of spectral markers**. Measuring the dependence of the precise abundance of a single analyte on physiological conditions in humans (e.g., given protein in states A, B, C) is known to be notoriously challenging. It is even more difficult to quantify concurrent changes in the abundance of many different substances belonging to distinct molecular classes (e.g., lipids and proteins) in a single experiment. Owing to its cross-molecular coverage, broadband infrared spectroscopy is able to make a valuable contribution here. Relative concentration changes of different molecular classes, in comparison with each other, can be estimated from the relative change in the ratio of the intensity of absorption bands, which are dominated by specific molecular classes and can therefore be assigned to them[23,34,35,37,43–53]. Table 1 shows a selection of peak ratios (together with their respective assignments) previously proposed as markers for physiological states, disease diagnostics, and monitoring.

We analyzed the within- and between-person variability of these ratios and evaluated their Index of Individuality (*II*), which is defined as the ratio of the average within-person variability $S_W$ and between-person variability $S_B$[9,10]:

$$II = S_W/S_B \qquad (1)$$

When a molecular marker has an $II < 0.6$, it is considered to be

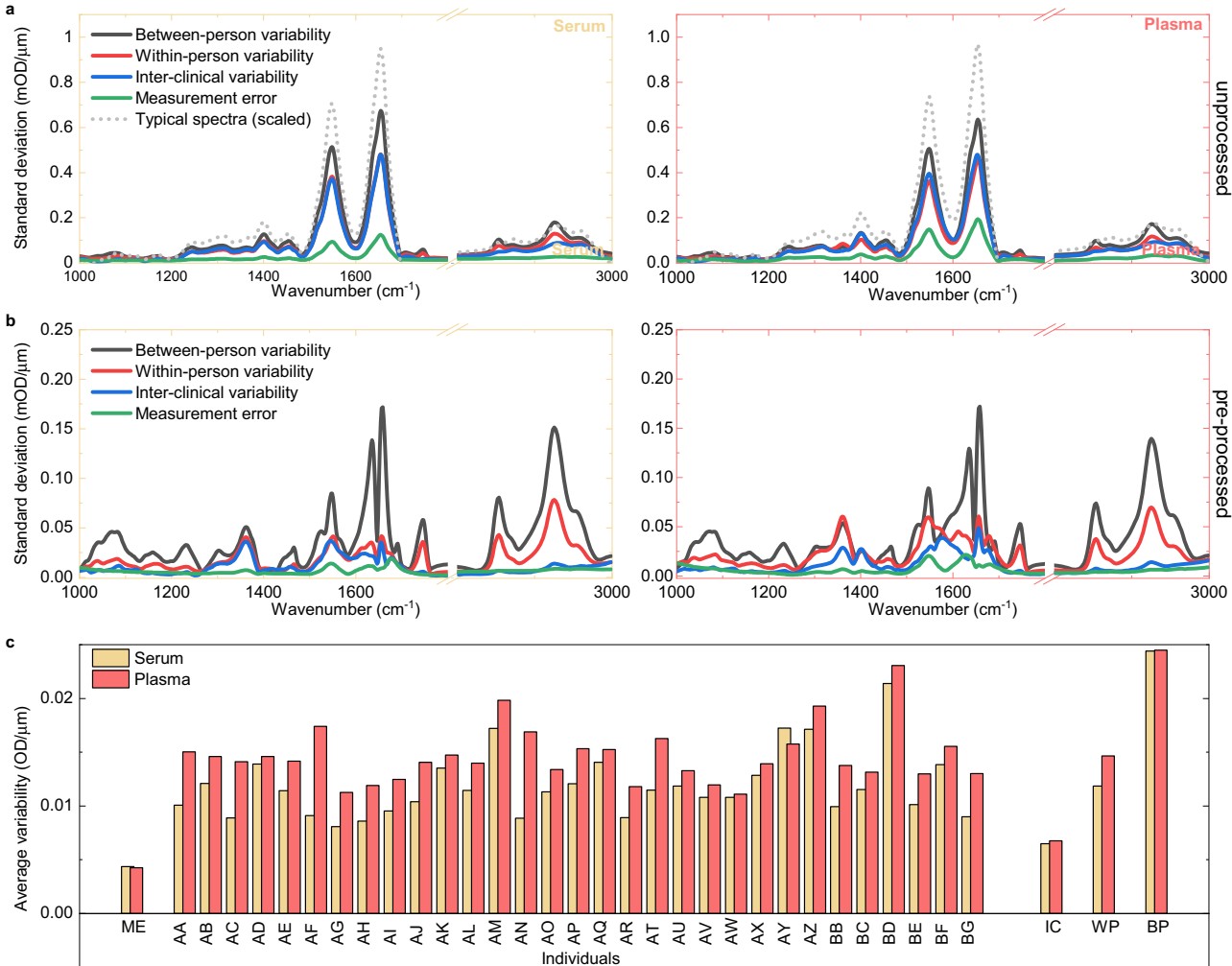

**Fig. 2 Biological variability of IMFs.** Spectrally resolved variability of **a** unprocessed and **b** pre-processed IR fingerprints obtained from serum and plasma samples: Standard deviation of the measurement error (green line), between-person (gray line), within-person (red line), and inter-clinical variability (blue line) are plotted against wavenumber. **c** Total variability of pre-processed spectra integrated over the whole spectral range (*ME* measurement error, *IC* inter-clinical variability, *WP* within-person variability, *BP* between-person variability).

specific to an individual[9] and can in principle be used to track the physiological state of an individual. In the context of disease detection, a low *II* also implies that the level of a marker may be within the normal range for one person, while the same values might be abnormal for another individual. In case–control scenarios, this may lead to deviations being erroneously identified as anomalies, which underlines the importance of quantitatively evaluating both the within- and between-person variability[9,10].

As a case example, the ratio of the intensity of the amide I main peak to that of its shoulder ($I_{1635}/I_{1654}$—Fig. 3a) contains information about the relative amounts of alpha-helix and beta-sheet structures[34,45,54]. This parameter was used in classical case-control studies[44,45] as well as for longitudinal disease monitoring[34,35]. We found its within-person variation to be up to five times smaller than the variation between subjects, as reflected in its low *II* of 0.23 and 0.27 for serum and plasma, respectively. The high degree of individuality of this ratio suggests that it may be most helpful in health monitoring, as IMFs are to be referenced to those previously acquired from the same individual.

Generally, we find that intensity ratios of plasma and serum spectra behave in a similar fashion over time, which emphasizes the fact that serum and plasma share the vast majority of their

molecular components and therefore provide similar information. We found that most of the peak ratios are rather stable over time, whereas for some individuals (e.g., $I_{1635}/I_{1654}$ of the subject BD, Fig. 3b, d) we observed a significant change over time. This shows on the one hand that these spectral markers can be measured reliably and stably over time, but also that changes of these markers over time (potentially due to a disease) can be detected. In general, many peak ratios are found to have a rather low Index of Individuality (Table 1), which makes their biological variability comparable to commonly measured clinical variables[55]. This connects IMFs to other analytical approaches and highlights its value as a source of highly specific and individual molecular information.

**Identification of person-specific IMFs in the liquid phase of human blood.** Several spectral features of the IMFs are found to exhibit a low Index of Individuality, which renders them highly person-specific. This raises the intriguing question whether IMFs permit identification of individual molecular phenotypes, despite the inevitable background of biological variability. Although NMR and mass-spectroscopic fingerprints of human urine[56], saliva[57], blood serum[58], and plasma[59] were found to possess this

**Table 1 Selected IR peak ratios with their assignments to respective physiological conditions.**

| Peak ratio | Serum | | Plasma | | Assignment/applications |
|---|---|---|---|---|---|
| | Value ± $S_B^2$ | II | Value ± $S_B^2$ | II | |
| $I_{1635}/I_{1654}$ | 0.756 ± 0.012 | 0.23 | 0.771 ± 0.012 | 0.27 | Ratio of β-sheet to α-helix secondary structures;[34,35,45,49] proposed marker for colitis;[35] determination of albumin-to-globulin ratio;[44] indicator of lymphoma and melanoma in a mouse model[45] |
| $I_{1546}/I_{1655}$ | 0.635 ± 0.003 | 0.61 | 0.638 ± 0.003 | 1.2 | Amide I to amide II ratio;[37,46,49,51] alternation of secondary structure;[46,51] formation of protein fibrils[46] |
| $I_{1655}/(I_{1655}+I_{1548})$ | 0.610 ± 0.001 | 0.63 | 0.608 ± 0.001 | 1.2 | Ratio of α-helix structure to total proteins[47] |
| $I_{1684}/(I_{1655}+I_{1548})$ | 0.213 ± 0.002 | 0.48 | 0.218 ± 0.002 | 0.69 | Ratio of antiparallel β-sheet structure to total proteins[47] |
| $I_{1515}/(I_{1655}+I_{1548})$ | 0.174 ± 0.002 | 0.27 | 0.175 ± 0.002 | 0.34 | Ratio of tyrosine-rich proteins to total proteins[47] |
| $I_{2959}/I_{2931}$ | 0.993 ± 0.020 | 0.52 | 1.005 ± 0.019 | 0.50 | $\nu_{as}(CH_3)$-to-$\nu_{as}(CH_2)$ ratio; length of lipid chains;[51] correlates with gastric cancer[51] |
| $(I_{2855}+I_{2927})/(I_{2962}+I_{2871})$ | 0.952 ± 0.023 | 0.55 | 0.942 ± 0.021 | 0.52 | Elongation of fatty acids;[34,46] correlates with breast cancer progression[34] |
| $(I_{2851}+I_{2927})/(I_{1655}+I_{1548})$ | 0.178 ± 0.007 | 0.48 | 0.179 ± 0.007 | 0.46 | Lipid-to-protein ratio[47] |
| $I_{1239}/(I_{2851}+I_{2927})$ | 0.424 ± 0.013 | 0.54 | 0.422 ± 0.011 | 0.52 | Ratio of phospholipids to total lipids[47] |
| $I_{1741}/I_{1640}$ | 0.029 ± 0.003 | 0.59 | 0.029 ± 0.002 | 0.57 | Lipid-to-protein ratio;[52] correlation with apoptotic cells[52] |
| $I_{1740}/I_{1400}$ | 0.118 ± 0.012 | 0.59 | 0.107 ± 0.009 | 0.59 | Lipid-to-protein ratio;[37,52] correlation with tumor progression in tissues[52] |
| $I_{2852}/I_{1400}$ | 0.500 ± 0.018 | 0.54 | 0.454 ± 0.014 | 0.56 | Lipid-to-protein ratio[37] |
| $I_{1450}/I_{1539}$ | 0.287 ± 0.003 | 0.30 | 0.297 ± 0.003 | 0.34 | Lipid-to-protein ratio[23] |
| $I_{1240}/I_{1517}$ | 0.408 ± 0.004 | 0.51 | 0.407 ± 0.004 | 0.57 | Degree of phosphorylation of tyrosine[46] |
| $I_{1045}/I_{1545}$ | 0.109 ± 0.003 | 0.41 | 0.109 ± 0.003 | 0.48 | Phosphate-to-carbohydrate ratio[23] |
| $I_{1080}/I_{1550}$ | 0.145 ± 0.004 | 0.40 | 0.143 ± 0.004 | 0.44 | Phosphate-to-amide II ratio[37,49,51] |
| $I_{1060}/I_{1230}$ | 0.705 ± 0.013 | 0.43 | 0.697 ± 0.012 | 0.55 | $\nu_s(PO_2^-)$-to-$\nu_{as}(PO_2^-)$ ratio[23] |
| $I_{1170}/I_{1080}$ | 0.905 ± 0.017 | 0.50 | 0.918 ± 0.016 | 0.54 | Relative content of nucleic acids; distinguishes sera of lung cancer patients from those of healthy individuals[49] |
| $I_{1030}/I_{1080}$ | 0.626 ± 0.006 | 0.81 | 0.636 ± 0.005 | 1.01 | Glycogen/phosphate ratio; indicator of metabolic turnover in cells[43,48,53] |
| $I_{1080}/I_{1243}$ | 0.726 ± 0.014 | 0.46 | 0.717 ± 0.013 | 0.56 | $\nu_s(PO_2^-)$-to-$\nu_{as}(PO_2^-)$ ratio[49] |
| $I_{1587}/(I_{1655}+I_{1548})$ | 0.145 ± 0.002 | 0.45 | 0.178 ± 0.002 | 0.56 | Ratio of free amino acids to proteins[47] |
| $I_{1156}/I_{1171}$ | 0.894 ± 0.007 | 0.45 | 0.898 ± 0.006 | 0.49 | Change of carbohydrate moieties in plasma globulins; correlates with Alzheimer's disease[50] |
| $I_{1243}/I_{1314}$ | 0.856 ± 0.013 | 0.41 | 0.802 ± 0.013 | 0.56 | Reflects changes in protein and nucleic acid levels[51] |
| $I_{1453}/I_{1400}$ | 0.801 ± 0.006 | 0.43 | 0.737 ± 0.007 | 0.61 | $\delta_{as}(CH_3)$-to-$\delta_s(CH_3)$ ratio[51] |

$S_B^2$ between-person variability, II index of Individuality, $\nu$ stretching, $\delta$ bending, s symmetric, as asymmetric vibrations.

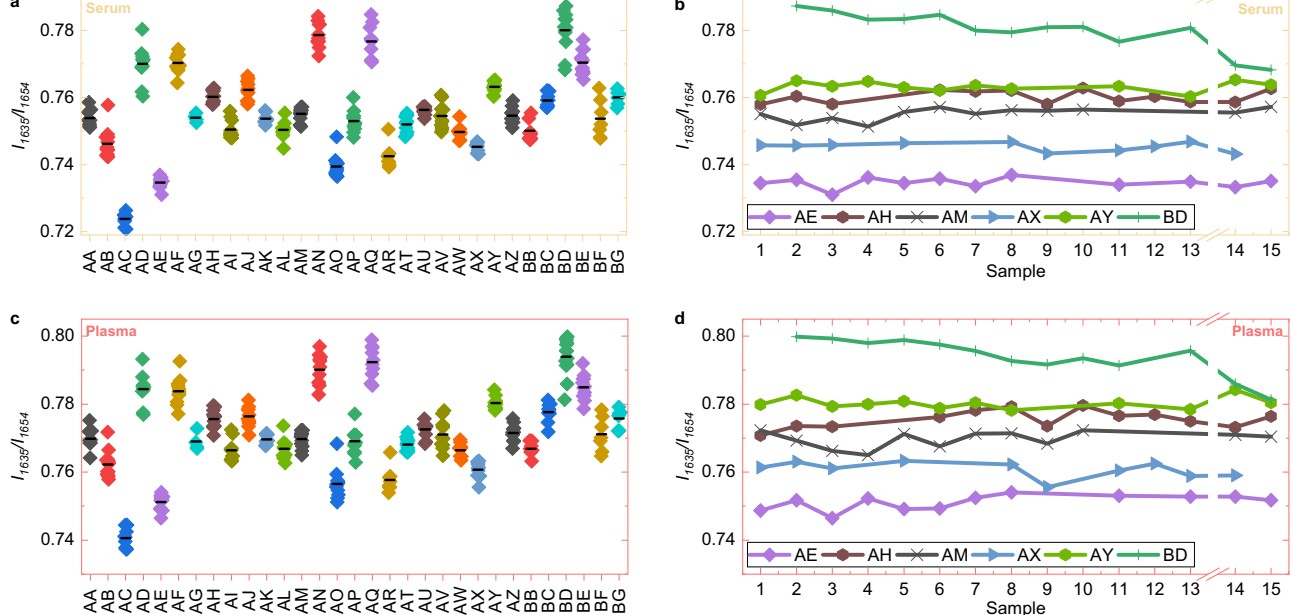

**Fig. 3 Stability of blood-based infrared spectral markers over time.** Ratios between amide I main and side peak ($I_{1635}/I_{1654}$) for **a**, **c** all individuals and for **b**, **d** six selected individuals representing detectable patterns of changes over time are shown. For most of the participants, the $I_{1635}/I_{1654}$-ratio was stable over time, while the pattern for e.g., subject BD (dark green line) was different over time. Each data point represents one blood sample. Blood samples 14 and 15 were collected 6 months after the previous ones.

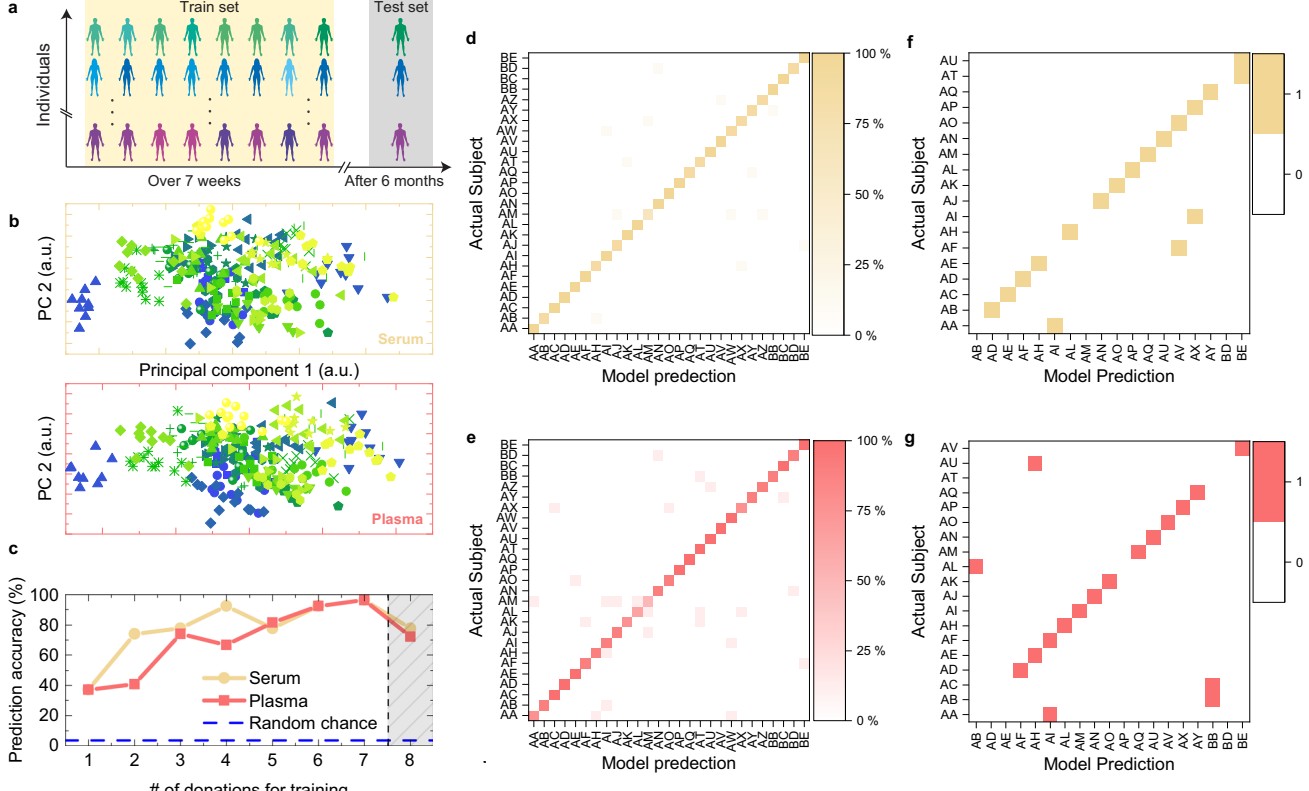

**Fig. 4 Identification of person-specific IMFs. a** Experimental setup with training and test sets used for comparative analysis of IR fingerprints (see also Supplementary Table 2). **b** Prediction accuracy of the random-forest classifier. The accuracy increases as more blood samples are used for training. When applied to IMFs that were sampled 6 months after the training IMFs, the accuracy of the classifier drops to slightly below 80% (shaded area). **c** Principal component analysis (PC1 versus PC2) of the 293 normalized IR spectra (for serum and plasma each) of the 27 individuals that were included in the machine-learning evaluation. All IMFs of one person are depicted by the same color and symbol. **d, e** Confusion matrix of the prediction density of a random-forest classifier, for each individual, using seven blood draws for training and one for testing and eightfold cross-validation. **f, g** Confusion matrix of the prediction accuracy for blood draws obtained 6 months after the initial sampling. The random-forest classifier was trained on the first eight blood samples (initial sampling). Serum results are displayed in orange **d, f** and plasma results in red **e, g**.

capability, analogous evidence for infrared fingerprints of human biofluids is lacking. To assess the existence of highly personalized IMFs, we examined the IMFs from participants who all provided blood samples at least eight times within the first 7 weeks of the sampling period in more detail (Supplementary Table 2). Using a descriptive investigation—with principal component analysis (PCA) of all 293 IMFs (for serum and plasma each) of the 27 individuals—we found that the infrared spectra of certain subjects can be readily distinguished, whereas others overlap significantly (Fig. 4c). The separation can be improved when higher principal components (PCs) are included in the analysis; however, perfect separation of all cases was not attained. Although PCs depict the maximum variance, these are not necessarily the "directions" in multi-dimensional space of IMF spectral amplitudes that maximize the inter-group separation[56].

Applying a random-forest machine-learning algorithm[60] (similar results can be also obtained with k-nearest neighbors[61] and XGBoost[62], Supplementary Figure 3 and Methods), we performed predictive analysis to derive classification models. The data for the first $N$ blood samples (for $N = 1\ldots 8$) per individual were used for training, and these classifiers were then tested on the data obtained from the following blood draw, $N + 1$ (Fig. 4a). The accuracy of the prediction is shown in Fig. 4c. If the classifier were predicting randomly, the accuracy would be 3.7%, as data from 27 participants were used in the training step. We show that training the algorithm with data from seven blood draws each, results in a prediction accuracy of >96%. Figure 4d, e shows the

result of a prediction-error analysis of a random-forest-based classification model for all individuals when seven blood draws per participant were used for training and one for testing, and when the data were subjected to eightfold cross-validation (repeated eight times with different combinations of training/test sets).

We observe that the vast majority of predictions lie on the diagonal of the confusion matrix (Fig. 4d, e), which demonstrates that the classifier is highly accurate, independently of the selection of the training set. This suggests the existence of highly person-specific IMFs that reflect the molecular phenotypes of individual donors, which are highly stable and reproducible over several independent blood draws (at least over 6 weeks). Investigation of the features that primarily contribute to successful classification revealed that the peaks that exhibit high-levels of between-person variation (e.g., 1747 cm⁻¹, 2854 cm⁻¹, 2929 cm⁻¹—mostly lipid absorption) most extensively contributed to the uniqueness of a person-specific IR fingerprints (Supplementary Figure 4).

In addition, we tested the possibility of deriving a multiclass classification model based on the intensity ratios (Table 1), again by using the random forests algorithm. We found that the average accuracy of the classifiers was 85% for serum and 75% for plasma. The full list of results—for both training and test sets—is provided in Supplementary Table 4. This outcome implies that intensity ratios capture a large fraction—but not all—of the relevant information contained in the spectra, thus highlighting the need for broadband infrared coverage. Notably, intensity

ratios ranked as highly important by this independent classifier coincide with those having a low Index of Individuality (Supplementary Table 3). It should also be noted that a fraction of these intensity ratios can be categorized as redundant, as many of them are highly correlated with others and thus provide no significant information gains (Supplementary Tables 5 and 6)—in contrast to the PCs which are by definition uncorrelated.

**Testing the long-term stability of infrared fingerprints**. Finally, we investigated the stability of IMFs on medically relevant timescales. Here, the prediction accuracy of sera and plasma sampled 6 months after the initial training sampling period was evaluated. Most of the individuals were still classified correctly. The number of misclassifications increased after 6 months, reducing the identification accuracy to ~80% (Fig. 4f, g). Considering the fact that part of the misclassification may have been caused by changes in the overall physiological states (e.g., lifestyle, new drug intake) of some of the subjects, which have not been investigated for this study, the overall chemical composition of human blood is remarkably stable even over a half a year. This finding emphasizes the method's suitability for health monitoring.

## Discussion

This proof-of-concept study demonstrates that (1) IMFs are robustly and directly measurable in liquid blood samples in a time- and cost-effective manner, (2) a single vibrational spectroscopic measurement provides access to multiple person-specific markers, and (3) infrared molecular phenotypes can be captured and monitored over time. Taken together, these findings suggest the possible applicability of blood-based infrared spectral fingerprinting for clinical health monitoring.

Routine blood profiling often focuses on the detection of defined analytes (e.g., molecule- or gene-based). However, broadband vibrational spectroscopy has the capacity to capture signals from all classes of biomolecular species. Thus, changes in any types of biomolecules, metabolic reaction products, or enzyme activities in human blood (e.g., elicited by a transition in health status) may lead to a change in the molecular phenotype of blood that may be reflected in the individual's IMFs. If so, regular, repeated sampling should enable any "abnormal" deviation in a molecular phenotype to be effectively detected by comparisons with previously recorded IMFs obtained from the same subject (self-referencing). In addition, any infrared measurement could represent a useful extension to current blood-based analytics, and could be followed up by well-established analytical approaches for deeper understanding. However, for the role proposed for IMFs in heath monitoring here, it is not necessary to understand the molecular origins of changes in IMFs, as long the characteristic deviation is specific and significant enough relative to its natural biological variability.

Although FTIR technology has been employed for case–control studies using dried serum and plasma samples[23–30], its applicability for human health monitoring has not been previously evaluated. Here, we applied FTIR to native liquid samples in a longitudinal study setting, and followed healthy, non-symptomatic individuals in order to quantitatively evaluate variations in the IMFs over time. We have shown that well-defined blood collection and processing workflows yield IMFs with a high degree of reproducibility, which allows cross-comparability across different clinical sites. Importantly, we find that the relative variations detected in IMFs are comparable to the variability of molecular concentrations measured with conventional analytical methods[63]. Furthermore, we demonstrate that many infrared spectral markers exhibit Indices of Individuality lower than 0.6,

placing them within the range of variability typically found for blood analytes routinely used in diagnostic medical laboratory facilities[55]. This demonstrates the ability of infrared fingerprinting to obtain highly person-specific information. More generally, our findings lay the foundation for a robust assessment of the existence of disease-specific infrared spectral features for health monitoring and disease detection.

Sampling individuals repeatedly over time, as we did here, can greatly enhance the capacity of infrared phenotypes to identify relevant information by eliminating the influence of day-to-day, within-person biological variability. In addition, any molecular phenotype may be more accurately detected in the context of longitudinal studies with self-referencing[2]. Such an approach will also eliminate the major source of "biological noise", namely between-person variability. This might be especially useful for diseases with the highest mortality rates (e.g., cancer, cardiovascular conditions), which often develop over the course of years or even decades, and where self-referencing based on IMFs could be particularly valuable. However, the answer to the question whether particular infrared spectral changes can be definitively linked to the onset or progression of a given disease is beyond the scope of the current work. For this purpose, sufficiently large cohort strata combined with clinical information are needed.

The data reported here show that the infrared molecular phenotype of an individual can be effectively followed over time. This is an essential prerequisite for future health monitoring and detection of medical phenotypes by infrared broadband vibrational spectroscopy, circumventing the need for any a priori knowledge about the molecular identity or causal origin of deviations from the normal physiological range.

## Methods

**Enrollment of study participants and blood sampling**. The study was reviewed and approved by "Ethikkommission bei der LMU München" (EK 20170820—Nr.: 17-532), and was conducted according to Good Clinical Practice (ICH-GCP), the principles of the Declaration of Helsinki, and all applicable legislations and regulations. Informed consent was obtained from all participants prior to blood collection.

Prior to the study, a statistical power calculation was performed to determine the sample size required to assess the mean and variance of the IMFs within a certain bound on accuracy, assuming a normal distribution. For the determination of the mean value of IMFs over all individuals at each wavenumber, a bound on accuracy of 0.025 mOD/μm with 95% confidence was set, resulting in a required minimum sample size of 26 individuals. The actual precision for the estimation of the mean is much higher than the stated limit, as several measurements per person were made and used for the analysis of the mean. For the estimation of the variabilities (between-person and within-person variability) we assumed that 26 persons donated 10 times. Thus, corresponding variabilities can be estimated within 35% of their true values and this can be achieved with a 95% confidence. To account for possible drop-outs over the course of the study, >30 individuals (31 individuals in total) were recruited.

Before each blood withdrawal, the participants were questioned about their health status and previous meals. Thirty-one adults were recruited for the longitudinal study and fasting blood samples were collected at the same site throughout the study. Within the first 7 weeks of the study, blood was sampled every 3–4 days. After 6 months, each participant again gave their blood two more times. Thus, each participant provided up to 15 samples over a period of >6 months (see also Supplementary Table 2). Ages of the participants ranged from 20 to 71 years with a mean of 39.6 years (±14.0 years, STD). 54.5% of the participants were female. None of the participants had any overt symptoms or severe diseases. Some of the participants were overweight, had allergies, food intolerances, or hypertension, which are all typical for a cross-section of the population at large (see Supplementary Table 1 for detailed information on subjects).

To evaluate the inter-clinical variability, five individuals volunteered to take part in an additional separate experiment. They gave blood at four different clinical sites within 4 h. No food was consumed during this time or within 6 h prior to the first sampling time.

**Standard operating procedures for serum and plasma sample collection, preparation, and storage**. Blood samples were collected, processed, and stored using defined standard operating procedures. Fasting blood was obtained between 9 am and 2 pm using Safety-Multifly needles of 21 G (Sarstedt), and transferred to

4.9-ml serum and 4.9-ml plasma Monovettes (Sarstedt). Special care was taken to make the different blood collections as comparable as possible. This meant that the same type of cannula was always used, the tubes were always filled to the recommended maximum filling level, serum was always collected first, and then plasma. For the blood clotting process to take place, the tubes were stored upright for at least 20 min and then centrifuged at 2,000 $g$ for 10 min at 20 °C. The supernatant was carefully aliquoted and frozen at −80 °C within 3 h after collection.

After all, samples were collected, the aliquots used for the actual FTIR measurement were prepared. One tube out of each of the smaller tube-sets was thawed and again centrifuged for 10 min at 2000 $g$. The supernatant was distributed into the measurement tubes (50 μl per tube) to be refrozen at −80 °C. All the FTIR measurements were performed upon two freeze–thaw cycles within the same measurement campaign.

To measure experimental errors during the experiment, quality-control (QC) samples from pooled human serum (BioWest, Nuaillé, France) were used[64].

**FTIR measurements**. The samples were measured in random order to reduce systematic effects. The spectroscopic measurements were performed in liquid phase with an automated FTIR device (MIRA-Analyzer, micro-biolytics GmbH) with a flow-through transmission cuvette (CaF$_2$ with ~8 μm path length). The spectra were acquired with a resolution of 4 cm$^{-1}$ in a spectral range between 950 cm$^{-1}$ and 3050 cm$^{-1}$. Note, that in comparison with measurements of dried serum, strong water absorption hinders the recording of the spectra over the entire mid-infrared range spanning from 400 cm$^{-1}$ to 4000 cm$^{-1}$ (see Supplementary Figure 1). After sample exchange, a water reference spectrum was measured to reconstruct the IR absorption spectra. After every five samples, a QC measurement was performed. Each measurement sequence usually contained up to 40 samples resulting in measurement times of up to 3 h. Experiments on QC showed that the change in IMFs of serum and plasma is negligible for the time span of a regular measurement sequence.

**Pre-processing of infrared absorption spectra**. All spectra were grouped according to the respective measurement day. The measured QC spectra of the different measurement days were compared with identifying small instrument drifts, and all the other spectra were corrected accordingly. "Negative" absorption, which occurs if the hydrated sample contains less water than the reference (pure water), was corrected for[65]. It is known from measurements of dried serum or plasma, that there is no significant absorption in the wavenumber region 1850–2300 cm$^{-1}$, resulting in a flat absorption baseline. We used this fact as a criterion for adding to each spectrum a water absorption spectrum taken from literature[66] to account for the missing water in the sample measurement and minimize the average slope in this region in order to obtain a flat baseline (Supplementary Figure 1). The same wavenumber region was subsequently utilized to compensate for baseline drifts, and all spectra were truncated to 1000–3000 cm$^{-1}$. Finally, all spectra were normalized as vectors, using Euclidean (or $L_2$) norm. To avoid $y$ axis scale change caused by Euclidean normalization, we computed average differences between maximum and minimum values of all spectra before normalization and then rescaled all normalized spectra to restore this averaged difference. This allowed us to preserve the average swing of the spectra and to correctly compare the variabilities of the pre-processed spectra.

**Evaluation of between- and within-person variability**. To obtain the within-person variability, we calculated the standard deviation of the participants' spectra over time and used all individual standard deviations to calculate the mean of the within-person variability. Between-person variability was obtained by averaging all spectra of a given individual and then calculating the standard deviation of these averaged spectra from different individuals. The inter-clinical variability was calculated in a similar manner from blood samples collected at different clinical sites and with standard deviations averaged to obtain the mean inter-clinical variation. The analytical error was estimated by repeatedly measuring quality-control serum samples and calculating the reproducibility of the obtained infrared spectra.

**Machine-learning analysis and classification**. After all samples and subject-related data were collected, the two following criteria led to the decision of the subset of 27 individuals to be considered in the machine-learning analysis:

1. Include at least 26 individuals (see also sample size calculation).
2. Include as many donations per individual as possible.

To meet both the above-listed criteria, we included only participants who have provided blood samples at least eight times within the first 7 weeks of the sampling period for the analysis of person-specific IMFs.

To reduce the dimension of data sets and explain the variance with a small number of linearly uncorrelated variables—PCs—we used PCA. When a significant fraction of the total variance is captured by the first two PCs, the separation between different classes can be conveniently represented by 2D scatter plots. As PCA is unsupervised, it is often used as the first analysis applied to a new data set[41].

For the derivation of classification models, we used Scikit-Learn[67] (v. 0.20.3), an open-source machine-learning framework in Python (v.3.6.8). We trained various models based on three algorithms: Random forests[60], k-Nearest-Neighbors[61], and XGBoost[62]. The purpose of classification is to predict and test the identity of individuals using multiclass classification models. It turned out that a random-forest-based model (an ensemble of 3160 decision trees) provided the highest accuracy. The prediction accuracy is defined as the proportion of individuals who are correctly classified according to the model applied. Information on the optimal values of model parameters can be found in the SI (caption of Supplementary Figure 3). The search for the optimal hyperparameters was performed using grid-search. Performance evaluation was carried out using cross-validation and its visualization using the notion of the confusion matrix.

Owing to the high dimensionality of the spectral data, and the high degree of correlation among the original features, the machine-learning algorithms were not applied directly to original data but rather to features extracted from them. The following approaches to feature extraction were used:

a. dimensionality reduction using PCA (described above). Thereby, the PCs transformation was fit on the training set only and used to transform both training and test set. The minimum number of PCs required to preserve 99.9% of the explained variance was kept.
b. manual extraction of spectral-intensity ratios.

In addition, we have evaluated the relative importance of each feature by measuring how much the tree nodes that use a particular feature reduce the average impurity (Gini impurity[68]) across all trees in the ensemble. This quantity is known as the Gini importance[69,70]. Gini importance is a way to measure the relative importance of each feature (in this case, wavenumbers) with a model build using the random-forest algorithm. Intuitively, it measures how much the tree nodes, across all trees of a random forest, reduce class impurity on average. By average, it is meant a weighted average, where each node's weight is equal to the number of training examples that are associated with it.

**Reporting summary**. Further information on research design is available in the Nature Research Reporting Summary linked to this article.

## Data availability
The authors declare that the main data supporting the findings of this study are available within the article and in Supplementary Data 1. Any additional information, data, and the statistical-analysis code are available upon request.

## Code availability
The custom code used for the production of the results presented in this manuscript is stored in a persistent repository at the Leibniz Supercomputing Center of the Bavarian Academy of Sciences and Humanities (LRZ), located in Garching, Germany. The code can be only shared upon reasonable request, as its correct use depends on the settings of the experimental setup and the measuring device and should therefore be clarified with the authors.

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

## Acknowledgements

We thank Jacqueline Hermann, Katja Leitner, Sigrid Auweter, Daniel Meyer, Beate Rank, and Incinur Zellhuber for their help with this study. In particular, we acknowledge the efforts of many individuals who participated as volunteers in the clinical study reported here. We also want to thank Alexander Žigman Kohlmaier and Frank Fleischman for their feedback on the manuscript.

## Author contributions

M.H. and M.Ž. designed the research plan; M.Ž. and F.K. initiated and led the study plan; N.H. led the clinical study; M.H., M.B., performed the measurements; M.H., K.V.K., M.B., L.V., and M.T. analyzed the data; M.H., M.Ž., K.V.K., L.V., and F.K. wrote the paper.

## Funding

## Competing interests

The authors declare no competing interests.
