## [Peer Review File · Nature Communications]

Reviewers' comments:

Reviewer #1 (Remarks to the Author):

This is an interesting study but no more than an incremental step in the field of spectrochemical analyses of biofluids for clinical applications. Even in a more specialised journal, it would really be a baseline for a further analyses. What is the basis of this healthy population - are they representative of a general population; one would need many more subjects to claim this. Whilst there is nothing fundamentally wrong with the experimental part of this study, I cannot believe this warrants publication in a Nature journal - apologies. Maybe Analyst or Analytical Methods would be a better target?

Reviewer #2 (Remarks to the Author):

The authors present a very interesting and timely study. However I cannot recommend publication of the study in its current form for 2 major reasons

1) The study highlights the use of liquid based serum spectroscopy and its potential benefits. However the raw spectra that have been reproduced by the authors do not appear to be liquid but rather appear to be dry spectra. The water overtone peak in the 2100 wavenumber region and more importantly the H-O-H bending at around 1645 is clearly visible - I think the authors are using a water background but please see comments later on. The liquid spectrum that the authors are claiming to be liquid in Fig 1 b Fig 2 a (the legend states that these are raw spectra) I think have been modified (and are not properly raw spectra) or are dry spectra as you can clearly see 2 defined amide I and II peaks that are in the relevant ratios. Further reading of the paper clearly states that the authors have pre-processed these spectra using a water correction algorithm - please can the authors clarify if these are raw or processed spectra and which spectra were actually used

In addition, the time of experimentation stated by the authors and the path length stated leads to a volume of serum or plasma that would easily be dry or at least be in a dried state that renders their use of liquid spectra as not being possible. If the spectra aren't liquid then the entire paper is called into question as this is where the authors have placed a lot of emphasis and conclusions / novelty of this paper on liquid analysis. The drying time and profile of serum and plasma is well known and from my own analysis we see spectral changes occurring that are indicative of drying within 2 minutes.

2) Please can the authors qualify the number of samples they have used - why did they choose 31 (actually 27 for the main part of the paper) and is this enough in order to establish their main conclusion. - previous published work on molecular phenotyping was performed on 1200 people (<https://link.springer.com/article/10.1007/s11306-014-0707-1>)

Less Major comments

1) Please provide evidence for the increased reproducibility of liquid spectra compared to dried spectra for your analysis - you state that it is more reproducible but don't say by how much etc. If you are actually analysing liquid spectra did you dry them and collect them by the same methodology to measure this?

2) Pg 5 - please list the spectral regions in which the inter clinical and within person variability is comparable

3) Fig 2 - please show the pre-processing that was performed to create b as the peaks do not look like normal FTIR spectra - why has the Amide I split in two?

4) Plasma and serum reproducibility - please state RSD throughout the paper. Also please make sure that no drying occurs as this will have an impact due to the major protein differences between these two fluids

5) Please explain what is meant by a water reference spectrum - was this the background spectrum collected and the serum the sample spectrum. If so please show the impact of this on the spectral collection - when this has been performed previously people have published the full spectrum from (4000 - 450) so the entire water region at the high end can be analysed as well - as using a water reference spectrum can cause negative peaks in that region. The authors do not state the spectral range they have collect the FTIR spectra over and they state that they correct for negative peaks using a previous method - so they collect the spectra but a full spectrum is never shown in the paper.

Please find detailed answers to the reviewers' comments to our manuscript (# NCOMMS-20-02649) below:

Reviewer #1 (Remarks to the Author):

This is an interesting study but no more than an incremental step in the field of spectrochemical analyses of biofluids for clinical applications. Even in a more specialised journal, it would really be a baseline for a further analyses. what is the basis of this healthy population - are the representative of a general population; one would need many more subjects to claim this. Whilst there is nothing fundamentally wrong with the experimental part of this study, I cannot believe this warrants publication in a Nature journal - apologies. May be Analyst or Analytical Methods would be a better target?

We would like to thank the reviewer for her/his feedback. Unfortunately, these statements suggest that the reviewer did not understand the purpose and main conclusions of our study. In what follows, we try to make some clarifications prompted by the comments of the reviewer.

We find the following hard to understand: *"what is the basis of this healthy population - are the representative of a general population; one would need many more subjects to claim this"*. We are well aware that our study is not suited to make statements at populational level, and such a claim would be scientifically unfounded. We did not make such statements in the manuscript. As noted in multiple places in the manuscript as well as in the abstract, our study presents the *so far unreported proof-of-concept* that blood-based infrared (IR) fingerprints are *robust* enough to ensure the capacity and potential for future health and disease monitoring, both of which critically depend on how stable infrared vibrational fingerprints of individuals are over time. We reveal the fingerprint robustness over time and do not attain to populational profiling. We think that going into the molecular-genetic definition of healthy population, genetic diversity, deep phenotyping, goes way beyond this work.

In fact, the study cohort we chose here to explore the inherent dynamics of IR fingerprints over time is not smaller than conceptually comparable previous high quality studies reporting on the foundation of individuality and resilience of metabolic fingerprints in human biofluids (PNAS PMID: 18230739; J Proteome Res PMID: 26055080). These earlier studies were layed down on studying 22 and 12 individuals over time, respectively. And it is important to realize that the essential point of our work is the study over multiple time-points over a relevant time-scale (weeks/months). If the reviewer thinks that these findings are not new or not important for the future clinical applications of infrared molecular fingerprinting, we kindly ask to please refer to previous work presenting these findings and/or explain why she/he feels the findings are of little or no importance for the purpose mentioned, respectively.

"This is an interesting study but no more than an incremental step in the field of spectrochemical analyses of biofluids for clinical applications". Such a severe criticism is entirely unacceptable without being justified by detailed factual arguments and reference to published work documenting this assessment. What is/are the publications the reviewer sets our work in relation to, when diminutively call the reported advance only *"incremental"*? *"Incremental"* with respect to specifically what? In the absence of any element of a factual criticism referring to previous work, we feel we are fully justified to maintain all major claims of our manuscript. The questions we have addressed in our work have not been addressed before in any systematic study, or at least have not been published before, to our knowledge. We maintain that our paper lays the foundation for using vibrational infrared fingerprinting of biofluids to a broad range of applications in biomedicine and basic research, and that the implications and citation potential are rather large, given that whole fields are affected. Not last, our study pertains to a real-world problem and lays the foundation for translation to direct applicability, which should be of interest for a journal like *Nature Communications*.

"Whilst there is nothing fundamentally wrong with the experimental part of this study...".

We would like to thank the reviewer for attesting this.

Reviewer #2 (Remarks to the Author):

The authors present a very interesting and timley study. However I cant recommend publication of the study in its current for 2 major reasons

We would like to thank Reviewer #2 for carefully considering this manuscript and providing valuable and constructive feedback. Both "major reasons" are based on a misunderstanding, which we wish to clarify below. In addition, prompted by the comments of the reviewer, we also prepared a manuscript where we are highlighting mentioned points for an easier overview and added some comments for the Reviewer #2.

1) The study highlights the use of liquid based serum spectroscopy and it potential benefits. However the raw spectra that have been reproduced by the authors do no appear to be liquid but rather appear to be dry spectra. The water overtone peak in the 2100 wavenumber region and more importantly the H-O-H bending at around 1645 is clearly visible - I think the authors are using a water background but please see comments later on. The liquid spectrum that

the authors are claiming to be liquid in Fig1 b Fig 2 a (the legend states that these are raw spectra) I think have been modified (and are not properly raw spectra) or are dry spectra as you can clearly see 2 defined amide I and II peaks that are in the relevant ratios. Further reading of the paper clearly states that the authors have pre-processed these spectra using a water correction algorithm - please can the authors clarify is these are raw or processed spectra and which spectra were actually used.

We would like to note that there must be a misunderstanding – we make it clear again, that all the samples in the study have indeed been measured as fluid, hydrated samples. Here the passage where we state this explicitly - in the **Methods** section of **FTIR measurements**:

“... The spectroscopic measurements were performed with an automated FTIR device (MIRA-Analyzer, micro-bioanalytics GmbH) with a flow-through transmission cuvette (CaF₂ with ~8 μm path length)...”

Given the flow-through cuvette that was constantly filled with a fluid – either being sample, transport fluid, washing fluid, the samples in this setting could never dry out. Combining this approach with a semi-automated sample delivery is the very advance of our study from a methodological point of view.

We would also like to note that although this is the first study examining such a high number of native samples in a semi-automated way, we are not the first ones performing FTIR measurements on hydrated samples. Whilst the reviewer is right that the majority of FTIR spectrometry is performed on dried samples, there are a few reports on comparable measurements and thus comparable infrared spectra:

- Here an example demonstrating measurements of fluid phase samples with cuvette of 7 micrometers thickness: Bunaciu, A. A. et. al.: *Vibrational Spectroscopy in Body Fluids Analysis. Crit. Rev. Anal. Chem.* 2017, 47 (1), 67–75. <https://doi.org/10.1080/10408347.2016.1209104>.
- A study discussing how to properly process infrared spectra of biofluids: Yang, H., et al.: *Obtaining Information about Protein Secondary Structures in Aqueous Solution Using Fourier Transform IR Spectroscopy. Nat. Protoc.* 2015, 10 (3), 382–396. <https://doi.org/10.1038/nprot.2015.024>.

The data in the literature listed above clearly demonstrate similarities to the nature of infrared spectra as displayed in our manuscript.

Moreover, our Figure 2a is indeed *not* displaying raw data, but rather standard deviation across infrared spectra. We would like to note that we have labelled the raw as well as processed data. Moreover, we have additionally highlighted the way we processed the data.

In addition, the time of experimentation stated by the authors and the path length stated leads to a volume of serum or plasma that would easily be dry or at least be in a dried state that renders their use of liquid spectra as not being possible. If the spectra aren't liquid then the entire paper is called into question as this is where the authors have placed a lot of emphasis and conclusions / novelty of this paper on liquid analysis. The drying time and profile of serum and plasma is well known and from my own analysis we see spectral changes occurring that are indicative of drying within 2 minutes.

As noted just above, the samples were indeed measured in their native liquid phase. Therefore, we need to stress once again that it is a misunderstanding to think they were dry. Importantly, we mentioned in the Methods section that we were using a fully closed fluidic system, to preserve the nature of serum and plasma as good as possible, as we are well aware of artifacts in measuring dried samples.

2) Please can the authors qualify the number of samples they have used - why did they choose 31 (actually 27 for the main part of the paper) and is this enough in order to establish their main conclusion. - previous published work on molecular phenotyping was performed on 1200 people (<https://link.springer.com/article/10.1007/s11306-014-0707-1>)

We would like to thank the reviewer for bringing our attention to the mass spectroscopic study of serum metabolome (Dunn et al. 2015, DOI: 10.1007/s11306-014-0707-1) where 1200 individuals were studied. However, the mentioned study provides an important reference dataset for understanding the “normal” variation in the human serum metabolome at a **single** time point, based on well-established analyses across molecular research and diagnostic laboratories.

Thus, this study is cross-sectional, and shows measurement of a *single time point*. It would be unfair to directly compare it to our study setup: Very specifically, we argue that it is not required for the impact of the results of our proof-of-concept study to be based on an extraordinarily large cohort size that the reviewer might possibly have had in mind when thinking of cross-sectional omics-profiling of blood samples (e.g. with mass spec or high-throughput sequencing). In fact, the study cohort size we chose here to explore the inherent dynamics of IR fingerprints over time is not smaller than conceptually comparable previous high-quality studies reporting on the foundation of individuality and resilience of metabolic fingerprints in human biofluids (PNAS PMID: 18230739; J Proteome Res PMID: 26055080). These earlier studies were laid down on studying 22 and 12 individuals over time, respectively. And it is important to realize that the essential point is the follow-up of individuals over multiple points over weeks and months, altogether encompassing more than 300 sampling points.

Thus, we think that our results may have major implications suggesting the possible applicability of blood-based infrared fingerprinting, not demonstrated on any human cohort previously with infrared fingerprinting.

Less Major comments

1) Please provide evidence for the increased reproducibility of liquid spectra compared to dried spectra for your analysis - you state that it is more reproducible but dont say by how much etc. If you are actually analysing liquid spectra did you dry them and collect them by teh same methodology to measure this?

As noted above, the measurements were performed in liquid phase. We never dried samples or measured dried samples, and think that including such measurements would be a deviation from the focus of our paper. Here we would like to specially highlight and recommend previously published study directly comparing reproducibility of dried serum versus liquid serum measurements (*Fabian H. et al, 2005 May-Jun;10(3):031103. PMID: 16229628*), where the authors conclude on the strength of the liquid sample technique - which allows IR spectra to be obtained in the conformation-sensitive amide I region with unprecedented reproducibility. For clarification to the reviewer we now made a reference to the mentioned study in the manuscript as well.

2) Pg 5 - please list the spectral regions in which the inter clinical and within person variability is comparable

We would like to thank the reviewer for highlighting that and have listed that for clarity in the manuscript.

3) Fig2 - please show the pre-processing that as performed to create b as the peaks do not look like normal FTIR spectra - why has the Amide I split in two?

We would like to note that this figure shows the standard deviation of the spectra. It demonstrates that after normalization most of the variation of the Amide I peak comes from the side peaks. We have highlighted it in the legend.

4) Plasma and serum reproducibility - please state RSD throughout the paper. Also please make sure that no drying occurs as this will have a impact due to the major protein differences between these two fluids.

As noted above, given that the cuvette as well as samples were hydrated at all times, this was an unfortunate misunderstanding that we hope to have cleared now.

5) Please explain what is meant by a water reference spectrum - was this the background spectrum collected and the serum the sample spectrum. If so please show the impact of this on the spectral collection - when this has been performed previously people have published the full spectrum from (4000 - 450) so the entire water region at the high end can be analysed as well - as using a water reference spectrum can cause negative peaks in that region. The authors do not state the spectral range they have collect the FTIR spectra over and they state that they correct for negative peaks using a previous method - so they collect the spectra but a full spectrum is never shown in the paper.

As noted in the Methods section, all measurements were performed with a commercial FTIR device specialized on the measurement of liquid samples. Unfortunately, the policy of the company is, that the device only provides the absorption spectra in a range from 930-3050 wavenumbers. The power spectra of the reference and sample measurement are not provided, and therefore we are unable to display them.

Internally, the device records a liquid water reference spectrum (power spectrum of MIR radiation after transmission through a cuvette filled with water) prior to a serum measurement (power spectrum of MIR radiation after transmission through a cuvette filled with liquid serum). Then, the absorption spectrum is calculated and provided to the user.

As the reviewer correctly pointed out, this procedure can cause negative absorption (as also visible in Fig. 1b). As described in the methods section, we apply a standard correction for this, as previously described by others. Another reason, why only the spectra between 930-3050 wavenumbers are recorded and shown, is that it is indeed from liquid serum. Strong water absorption bands below 3050 and above 930 wavenumbers attenuates the light too strongly to reconstruct a useable spectrum. Therefore, spectrum of liquids samples is often only shown between 1000-3000 wavenumbers.

REVIEWER COMMENTS

Reviewer #1 (Remarks to the Author):

The authors have completed and addressed the comments made by the Reviewers.

Reviewer #2 (Remarks to the Author):

Thanks to the authors for the response and I do understand that the measurements that have been undertaken have been undertaken as fluid samples. I thank the reviewer for updating the methodology and clarifying where possible. However can the authors provide evidence that the samples are still liquid as this has not been provided.

In addition the authors have not qualified why they chose the number of samples - why did they choose this number and what evidence do they have in order to justify stopping at this number of patients - for instance have they performed a power calculation or anything similar.

With regard to the water reference spectrum - the water reference spectrum is clearly used by the spectrometer and is an important component to understand the fact that the company does not display it only makes me wonder further as to what it is like and if it is so important to the study it should be possible to collect and display. In addition can the authors further explain the impact of using the correction on a limited wavenumber range.

Reviewer #3 (Remarks to the Author):

General Comments:

The authors present a new method to measure multiple molecular components of blood (plasma or serum) simultaneously using Infrared vibrational spectroscopy with Fourier-Transform Infrared (FTIR) spectroscopy. At the current time there is a large increase in the amount of data that researchers/clinicians can obtain on an individual. Methods that allow summarization of high dimensional data are sorely needed. This method allows researchers and clinicians to obtain infrared molecular fingerprints (IMF) on many hundreds (thousands) of circulating blood components that might be used to evaluate health status when measured over time. In this paper, the authors examine such measures in 31 healthy adults measured up to 13 times over the course of 7 weeks and then again 1-2 times at 6 months. With these measures the authors can assess single time differences between individuals, short-term changes within and between individuals and long-term changes within and between individuals. The data suggest that the IMFs are stable over time. The lower variability within a person compared to that between persons also enables researchers/clinicians to identify person-specific patterns over time. Thus, such data could be used to monitor health status of individuals in a timely and cost effective workflow. While I can see the benefits of this approach as presented in this paper, there are several issues that should be addressed by the authors to convince readers of the value of their new method.

1. The manuscript should be carefully reviewed to improve its language. There are a number of awkward sentences that should be modified. A few examples are
 - a. Line 5: "across a wide range ..."
 - b. Line 84: we usually say "in relation to" instead of "in relation with"
 - c. Line 142: it should state "here is better than what has been shown ..."
 - d. Line 161: "...higher variance levels..."
 - e. Line 174: "in dependence of" is very awkward and it is not clear what you mean.
2. The abstract states that the "variation of person-specific spectral markers over time is up to a factor of five lower than that across a cohort ..." However, this point is not really made in the paper itself. If true, the authors should highlight this point in the paper as well.

3. There is a great deal of missing information in the paper. Without such information it is difficult to judge the new method.

a. More descriptive information should be provided on the study participants. We are only told that they are “healthy” and are ages 20-71 with a mean age of 39.6 ± 14 years and 54.5% are female. But what makes them healthy? For example, what are their BMIs, diabetes rates, cardiovascular risks (e.g. blood pressure, lipid levels, blood pressure treatment, lipid treatment), cancer risks. No information is provided. A table could be added with this descriptive information.

b. It would be useful to have a figure that has a row for each subject indicating when they had measurements taken with time of measurement on the x-axis. If all 31 subjects had 13 measures during 7 weeks and 2 measures at 6 months, then there would be a total of 465 measurements. Figure 1a provides a crude picture of the times of measurements, but it should be more detailed, providing information on each of the 13 time points during the 7 weeks and the 2 time points at 6 months with each person’s trajectory over time. Note that this figure also refers to 6 weeks but the manuscript refers to 7 weeks.

c. It is unclear how many total observations were taken in this study and the total number should be stated. The only mention of the total number of observations is in the legend of Figure 1 where it states that there are 636 measurements from 31 individuals. This total suggests that there are almost 20 observations per subject, a number that is much higher than reported in the paper.

d. The authors report that multiple samples were taken at different clinics within 4 hours’ time in order to assess the effects of blood collection and sample handling processes. It is unclear how these additional measures were used in evaluating a person’s trajectory over time. One way would be to select one of the observations at that time and use it to evaluate trajectory over time. Probably it would be best to use the one from the clinic that the person usually attends. (Did subjects go to the same clinic for each measurement?) A second way would be to include clinic in the analysis model with repeated measures for those individuals who went to different clinics. There is no discussion of how these additional measures were dealt with in the analysis. Of course, clinic should be a factor in the model assessing effects over time as well.

4. Lines 179-181: The authors refer to “relative changes in concentration” on Page 7. I see that these are relative changes in intensity of absorption bands, but what are they relative to?

5. While the statistical analyses appear to be valid, there are many details of the analysis of the data that are missing.

a. What was the actual model used for assessing trends over time? Or was any model used? In “Evaluation of between- and within person variability” the authors only mention descriptive statistics of averaging standard deviations. There is no formal method described for evaluation these trends over time and across people. Typically, one would apply a linear mixed effects model.

b. The use of principal components analysis to obtain independent measures over molecular spectra makes sense; however, it is unclear exactly what the input for the PCA was. I expect that all observations for each individual were used. Is that correct? Or was it some summary measure(s) drawn from the observations for each person? Further, how these principal components were used? In particular, were these PCs used as input to the classification models? And how many PCs were used as input?

c. The description of the classification models is minimal. Were PCs for each person used as input? What are you trying to classify? Is the output the classification of who a person is? How did you define prediction accuracy? Is it the proportion of people who are correctly classified based on the algorithm?

d. The Gini importance values in Table S2 are small. Can you provide some interpretation of the Gini importance index for readers who may be unfamiliar with this measure?

e. For the “Evidence of person-specific IMFs” the authors use a subset of 27 individuals. Why a subset of the 31 subjects and how was this subset selected?

f. Figure 3: Six individuals were selected for Figures 3b and 3d. How were these individuals selected? These figures do a good job of showing that there are individual specific patterns over time, but we don’t know if these individuals were selected for this purpose or whether they were randomly selected.

6. The most convincing evidence that the authors can provide would be to compare their results for “healthy” individuals with those of some who are not healthy, though this goal may be beyond the scope of this paper.

Minor Comments

1. Line 88: I think the word “variability” would be better than “instability”.

2. Could a reference be included for the "Well-defined standardized protocols"
3. Figure 3 legend: The legend incorrectly identifies the I1548/I1654 ratio, but it appears to be the I1635/I1654 ratio.
4. Figure 4c: what do the different colors represent? What do the different symbols represent?
5. Figure 4d: The off-diagonal squares are so light that you cannot see them at first glance, especially in Figure 4d.
6. Figure 4f,g: Are the axes correctly labeled? It appears to me that none of these individuals were correctly classified, since the axes do not match. So how can these be proper confusion matrices?
7. Figure S2 has an X-axis from 1 to 7. Since subjects had up to 13 measures over 7 weeks, which subset of measures were selected for this figure? I realize it is the number of measure used to obtain the prediction for a person, but what subset of the 13 observations were used? Further, what are the values used for parameters for each classification method, even if they are the default values?
8. What do the colors mean in Table S1? What do the colors mean in Tables S3 and S4?

Dear Referees,

We would like to thank you for your thoughts and comments that have all helped us to improve our manuscript. In what follows, we address your comments point-by-point. We have enumerated the revisions (REV 1, REV 2, ...) made in compliance with your comments, and, for your convenience, indicated them in a marked copy of the revised manuscript, with all relevant changes highlighted. In addition, we added a version of the manuscript with tracked changes.

Response to Referee #1

The authors have completed and addressed the comments made by the Reviewers.

We are pleased that we were able to satisfactorily address all points of Referee #1.

Response to Referee #2

Thanks to the authors for the response and I do understand that the measurements that have been undertaken have been undertaken as fluid samples. I thank the reviewer for updating the methodology and clarifying where possible. However can the authors provide evidence that the samples are still liquid as this has not been provided.

We thank the Referee for appreciating our efforts explaining our measurement methodology of liquid samples. To explain the measurements on liquids in more detail, as well as to provide further proof that the samples are still liquid when measured, we have added a figure S1 to the Supplementary Information, which shows the acquisition of the spectra and the subsequent data processing in more detail (**Revision 1**). The figure displays the spectra of the light source through the measurement cuvette when no liquid is present and when the cuvette is filled with the sample/or water. This comparison clearly shows the difference in the spectra for liquid and non-liquid samples. The large differences between the spectra of dry and liquid samples would make it apparent when the sample is no longer liquid and can therefore be used as a control to ensure that the samples are still liquid during the measurement.

In addition the authors have not qualified why they chose the number of samples - why did they choose this number and what evidence do they have in order to justify stopping at this number of patients - for instance have they performed a power calculation or anything similar.

As proposed by the Referee, we have indeed performed a statistical power calculation to estimate the appropriate number of individuals to be included into the study. This was

performed when planning the clinical study. To give a better estimate to the Referee, the following passage is a direct excerpt from the Study Protocol that was submitted and approved by the ethics committee prior to the start of the study.

Please find a short excerpt of the relevant “*Statistical Power Calculations*” from the Study Protocol copied here:

“The sample size is determined to determine the mean and variance at single spectra values within a certain bound on accuracy. For data assumed to follow a Normal distribution, the bound or half-length of a 95% confidence interval (CI) for the mean is $2 \cdot \sigma / \sqrt{n}$, where σ is the population standard deviation and $\sqrt{}$ denotes the square root function. With $\text{bound} = 2 \cdot \sigma / \sqrt{n}$, setting bound to be the desired limit on accuracy, providing a prior estimate for σ and solving for n yields the desired sample size as $n = 4 \cdot \sigma^2 / \text{bound}^2$. An estimate for σ for healthy controls is provided by a pilot study performed on 36 cancer cases opposed to 36 healthy controls shown in Figure 1. This study focuses on the healthy controls, which have lower variability in addition to a lower mean curve. The point of highest variability occurs near wavelength 1075 where at least 95% of the individual observations appear to fall within a range of .1 centered at the mean. Setting $.1 = 4 \cdot \sigma$, the complete width of a 95% confidence interval for the spectra at a single point and solving for σ yields $\sigma = 1/40$. Setting the bound = .01 and $\sigma = 1/40$ yields a required sample size of 26 individuals as estimated from the sample size formula. The precision for estimating the mean will be higher than the bound specified because multiple measurements will be taken per individual and all used in the analysis of the mean. To account for potential drop-out or other technical problems, 30 individuals will be recruited. Variances require larger sample sizes for precise estimation but also require that all time points be used, which increases the precision. Therefore, we calculate precision obtainable with a minimal sample of 260 observations (10 measurements from 26 people). The observed precision will be much higher as this only accounts for the first wave and five waves are anticipated. Drop-out across waves are anticipated and this calculation protects against a worst case scenario where all individuals would drop out for all future waves. On the other hand, the sample size is calculated for estimating the total variability and in the analysis this estimate will be partitioned into the inter- and intra-individual components as well as the long-term dimension of time. Including all repeated samples at all waves for all individuals will ensure accurate precision for the three components of variability. A 95% CI for a population variance σ^2 is (L,U), with $L = (n-1)s^2 / q_{\text{chisq}}(.975, n-1)$, $U = (n-1)s^2 / q_{\text{chisq}}(.025, n-1)$, where $q_{\text{chisq}}(a, n-1)$ denotes the a-quantile of the chi-square distribution with $n-1$ degrees of freedom and s^2 denotes the sample variance. We set $U-L$ to be less than a proportion p of s^2 . This means we find n such that $(n-1)[1/q_{\text{chisq}}(.025, n-1) - 1/q_{\text{chisq}}(.975, n-1)] < p$. A sample of size 260 (10 time points for 26 participants) implies that the true variability can be estimated with precision 35% of its value.”

For clarification, we included a summarized version of the power calculation in the revised method section of the manuscript (**Revision 2**).

With regard to the water reference spectrum - the water reference spectrum is clearly used by the spectrometer and is an important component to understand the fact that the company does not display it only makes me wonder further as to what it is like and if it is so important to the study it should be possible to collect and display.

We agree with the Referee that access to the water reference spectra would be beneficial and it is regrettable that the company does not provide them. For future measurements we aim to gain access to the full spectral information by switching to another spectrometer. Although we are not able to provide the water references for the measurements presented in this work, we have performed additional measurements with another spectrometer using a similar measurement cuvette. This shows how the water reference spectra for such measurements would look in principle (**Revision 1**).

In addition can the authors further explain the impact of using the correction on a limited wavenumber range.

For clarity, we added a new Figure S1 to the Supplementary Information, which displays how the water correction algorithm works (**Revision 1**). This Figure further shows that strong water absorption above 3000 cm^{-1} , and the limited transmission of CaF_2 below 1000 cm^{-1} , reduces spectral intensity of the transmitted light to such a level, that it cannot be used for spectroscopy anymore. Therefore, the spectral range has to be limited from 1000 cm^{-1} to 3000 cm^{-1} , when performing spectroscopy on liquids.

In addition, we provide an explanation in the Methods section, explaining why the water correction algorithm only considers the wavenumber range $1850\text{-}2300\text{ cm}^{-1}$ as a correction criterion, and copy it below:

“It is known from measurements of dried serum or plasma, that there is no significant absorption in the wavenumber region $1850\text{-}2300\text{ cm}^{-1}$, resulting in a flat absorption baseline. We used this fact as a criterion to add to each spectrum a water absorption spectrum taken from literature⁶⁴ to account for the missing water in the sample measurement and minimize the average slope in this region in order to obtain a flat baseline (Fig. S1).”

Response to Referee #3

General Comments:

The authors present a new method to measure multiple molecular components of blood (plasma or serum) simultaneously using Infrared vibrational spectroscopy with Fourier-Transform Infrared (FTIR) spectroscopy. At the current time there is a large increase in the amount of data that researchers/clinicians can obtain on an individual. Methods that allow summarization of high dimensional data are sorely needed. This method allows researchers and clinicians to obtain infrared molecular fingerprints (IMF) on many hundreds (thousands) of circulating blood components that might be used to evaluate health status when measured over time. In this paper, the authors examine such measures in 31 healthy adults measured up to 13 times over the course of 7 weeks and then again 1-2 times at 6 months. With these measures the authors can assess single time differences between individuals, short-term changes within and between individuals and long-term changes within and between individuals. The data suggest that the IMFs are stable over time. The lower variability within a person compared to that between persons also enables researchers/clinicians to identify person-specific patterns over time. Thus, such data could be used to monitor health status of individuals in a timely and cost effective workflow. While I can see the benefits of this approach as presented in this paper, there are several issues that should be addressed by the authors to convince readers of the value of their new method.

We thank the Referee for appreciating our work presented in the manuscript and for sharing our view that infrared molecular fingerprinting is a cost-effective tool to access information from multiple molecular components of blood simultaneously, and thus presents a promising

new route to evaluate the health status of an individual (over time) in a minimally-invasive fashion.

We would especially like to thank the Referee for his detailed and constructive criticism, which has enabled us to further improve our work and shape our manuscript. In the following we have addressed all the points he/she raised:

- 1. The manuscript should be carefully reviewed to improve its language. There are a number of awkward sentences that should be modified. A few examples are**
 - a. Line 5: “across a wide range ...”
 - b. Line 84: we usually say “in relation to” instead of “in relation with”
 - c. Line 142: it should state “here is better than what has been shown ...”
 - d. Line 161: “...higher variance levels...”
 - e. Line 174: “in dependence of” is very awkward and it is not clear what you mean.

The language of the entire article was carefully reviewed by a language professional. The text passages addressed by the Referee were changed, and further linguistic improvements were made. All changes are indicated in a version of the revised manuscript with tracked changes.

- 2. The abstract states that the “variation of person-specific spectral markers over time is up to a factor of five lower than that across a cohort ...” However, this point is not really made in the paper itself. If true, the authors should highlight this point in the paper as well.**

We would like to thank the Referee for noting this. In response, we have gladly added a corresponding passage in the revised version of the manuscript to make the finding explicitly formulated (**Revision 3**).

- 3. There is a great deal of missing information in the paper. Without such information it is difficult to judge the new method.**
 - a. **More descriptive information should be provided on the study participants. We are only told that they are “healthy” and are ages 20-71 with a mean age of 39.6±14 years and 54.5% are female. But what makes them healthy? For example, what are their BMIs, diabetes rates, cardiovascular risks (e.g. blood pressure, lipid levels, blood pressure treatment, lipid treatment), cancer risks. No information is provided. A table could be added with this descriptive information.**

We agree with the Referee that it is indeed not trivial to define the quantitative level of “human healthiness” - there are human physiological abnormalities that may not induce symptoms and are also not detectable with current medical approaches. Therefore, we refrain from using the sole term “healthy” and instead use the phrase “healthy, non-symptomatic individuals” in the revised manuscript.

In addition, we followed the suggestions of the Referee and added Table S1 to the Supplementary Information (**Revision 4**). Here we list all potentially relevant information recorded on the volunteers in the study. Moreover, although perhaps not highlighted sufficiently previously, we would like to make clear that only the participants that did not have any diagnosed diseases nor did show any symptoms of any known diseases were enrolled into our study. This information was added to the Methods section (**Revision 4**).

- b. It would be useful to have a figure that has a row for each subject indicating when they had measurements taken with time of measurement on the x-axis. If all 31**

subjects had 13 measures during 7 weeks and 2 measures at 6 months, then there would be a total of 465 measurements. Figure 1a provides a crude picture of the times of measurements, but it should be more detailed, providing information on each of the 13 time points during the 7 weeks and the 2 time points at 6 months with each person's trajectory over time. Note that this figure also refers to 6 weeks but the manuscript refers to 7 weeks.

Here we would like to thank the Referee for the valuable suggestion and notion of our mistake in the Figure 1a. In the revised manuscript we changed the label to 7 weeks, and also added a more detailed overview of subject sampling in Table S2 of the Supplementary information (**Revision 5**). It shows the sampling time of all collected blood samples for all individuals. Also, the new table provides detailed information on the total number of samples per individual and the data used for the different evaluations.

c. It is unclear how many total observations were taken in this study and the total number should be stated. The only mention of the total number of observations is in the legend of Figure 1 where it states that there are 636 measurements from 31 individuals. This total suggests that there are almost 20 observations per subject, a number that is much higher than reported in the paper.

We would like to note that the exact number of 318 observations can now be taken from the newly introduced Table S2 (**Revision 5**). 636 measurements however arise from the fact that from each volunteer, at each observation time point, one serum, as well as one plasma sample, were collected and measured with FTIR. We have now clarified this imprecise information by adjusting the caption of the Figure 1 (**Revision 6**) and Table S2, of the revised manuscript (**Revision 5**).

d. The authors report that multiple samples were taken at different clinics within 4 hours' time in order to assess the effects of blood collection and sample handling processes. It is unclear how these additional measures were used in evaluating a person's trajectory over time. One way would be to select one of the observations at that time and use it to evaluate trajectory over time. Probably it would be best to use the one from the clinic that the person usually attends. (Did subjects go to the same clinic for each measurement?) A second way would be to include clinic in the analysis model with repeated measures for those individuals who went to different clinics. There is no discussion of how these additional measures were dealt with in the analysis. Of course, clinic should be a factor in the model assessing effects over time as well.

We believe that the Referee's comment is based on a misunderstanding, caused by an imprecise formulation on our part. The evaluation of the inter-clinical variability and the evolution of a person's trajectory over time were two separate experiments that are not directly connected. The main purpose of blood sampling at different clinics was to get a measure for potential (analytical) errors caused by sample collection, handling and preparation.

The aim was to examine and compare the extent of the inter-clinical (analytical) variability in relation to the within-person, biological variability. We reassuringly found that *the extent of variability introduced by sampling at different clinics is extensively lower than the average variability of a person over time*. This is an important confirmation of our precise and standardized sample collection and processing procedures that enables very reproducible sampling at different clinics by different personnel.

We agree with the Referee, that it would be indeed interesting to collect the samples of the participants in the longitudinal study also in a collection at different clinics, as it is closer to the daily clinical practice. It is however important to note that before our study it was not known what is the analytical error of blood-based infrared molecular fingerprints in the liquid phase, nor was it known what is the biological variability inherent to any given non-symptomatic human population. To assess both these independently, we designed a study enabling us to disentangle the two. Thus, in the current study, all the longitudinal blood collections of all participants were collected at the same centre on purpose - with the aim to quantitatively evaluate the noise possibly introduced by differences in sampling (analytical errors) from biological variability.

For bringing such an approach closer to clinical use, one will indeed need to additionally evaluate the level of variability within and between individuals when samples are collected at different clinics. Although we currently cannot combine these two experiments, we will take this suggestion into account when planning the next study.

For clarity, we stated in the Methods section that these are two separate experiments and that all blood samples of the longitudinal study were taken at one and the same place (**Revision 7**).

4. Lines 179-181: The authors refer to “relative changes in concentration” on Page 7. I see that these are relative changes in intensity of absorption bands, but what are they relative to?

We agree with the Referee and have thus changed the corresponding sentence in the revised manuscript into the following (**Revision 8**):

“Relative concentration changes of different molecular classes, in comparison to each other, can be estimated from the relative change in the ratio of the intensity of absorption bands, which are dominated by specific molecular classes and can therefore be assigned to them.”

5. While the statistical analyses appear to be valid, there are many details of the analysis of the data that are missing.

a. What was the actual model used for assessing trends over time? Or was any model used? In “Evaluation of between- and within person variability” the authors only mention descriptive statistics of averaging standard deviations. There is no formal method described for evaluation these trends over time and across people. Typically, one would apply a linear mixed effects model.

For the evaluation of variabilities, no statistical modelling was performed for assessing trends over time. Our approach is based on the estimation of the analytical, within-subject (intraindividual), and between-subject (interindividual) coefficients of variation and the derived index of individuality (a formal analytical-chemistry method), closely following related works such as Ref. 9 and 10, cited in the main text.

b. The use of principal components analysis to obtain independent measures over molecular spectra makes sense; however, it is unclear exactly what the input for the PCA was. I expect that all observations for each individual were used. Is that correct? Or was it some summary measure(s) drawn from the observations for each person? Further, how these principal components were used? In particular, were these PCs used as input to the classification models? And how many PCs were used as input?

For visualization purposes, such as in the Figure 4 c), all observations of the 27 individuals that were included in the machine learning algorithm were used. We have now added this information to the text for clarification (**Revision 9**).

In the case of classification, PCA was used as a way of reducing the dimensionality of the original space (wavenumbers) and extracting uncorrelated features which were then used as the input for the classification models. The principal components transformation was fit on the train set only and used to transform both, the train as well as the test set. In our study we always included the minimum number of principal components required to preserve the 99.9% of the explained variance. This information was added to the Method section of the revised manuscript (**Revision 10**).

c. The description of the classification models is minimal. Were PCs for each person used as input? What are you trying to classify? Is the output the classification of who a person is? How did you define prediction accuracy? Is it the proportion of people who are correctly classified based on the algorithm?

The idea is to predict the identity of individuals at each point in time, using multi-class classification models trained on the donations (for all individuals) previously collected. Towards that end, the principal components transformation was fit on all observations (of all 27 individuals) contained in the train set at each time point. The prediction accuracy was indeed defined as the proportion of individuals who are correctly classified.

To extend the explanation, we have added the above information to the Method section of the revised manuscript (**Revision 11**).

d. The Gini importance values in Table S2 are small. Can you provide some interpretation of the Gini importance index for readers who may be unfamiliar with this measure?

Gini importance is a way to measure the relative importance of each feature (in this case peak ratios) with a model build using the random-forest algorithm. Intuitively, it measures how much the tree nodes, across all trees of a random forest, reduce class impurity on average. By average, it is meant a weighted average, where each node's weight is equal to the number of training examples that are associated with it.

For better understanding, we added the above paragraph to the Method section and the following explanation to the caption of the corresponding Table S3. (**Revision 12**):

"It can be interpreted as a relative (and not absolute) measure of the importance of each feature and thus its values can be interpreted as high or low only relative to each other."

e. For the "Evidence of person-specific IMFs" the authors use a subset of 27 individuals. Why a subset of the 31 subjects and how was this subset selected?

According to our study protocol and the sample size calculation, the sampling of 26 individuals would have been sufficient for the purposes of our study. However, to account for potential drop-outs, at the initial time point of subject enrolment 31 subjects were recruited. For the "Evidence of person-specific IMFs", we investigated the person-specific information included in the IMFs by evaluating the accuracy of machine-learning models (trained using the IMFs) through time, longitudinally.

After all samples and subject-related data were collected, the two following criteria led to the decision of the subset of 27 individuals to be involved in the analysis:

1. Include at least 26 individuals (minimum required number according to the calculations in the study protocol);
2. Include as many donations per individual as possible.

Thus, we included only participants who have all provided blood samples at least 8 times within the first 7 weeks of the sampling period for the analysis of person-specific IMFs, to meet both the above-listed criteria.

For clarity, we have added a short passage on that in the Methods section of the revised manuscript (**Revision 13**).

f. Figure 3: Six individuals were selected for Figures 3b and 3d. How were these individuals selected? These figures do a good job of showing that there are individual specific patterns over time, but we don't know if these individuals were selected for this purpose or whether they were randomly selected.

These six individuals were selected on purpose, as they reflect the main findings based on the data of the entire studied cohort.

The reasons why we have chosen to showcase this specific subgroup are as follows:

1. All individuals in Figure 3 have donated at least 10 times and also participated in the second part of the campaign that took place 6 months after the initial sampling.
2. The I_{1548}/I_{1654} -ratio of the selected individuals does not overlap too much. This makes the figure easier to read and follow individual traces.
3. We wanted to show that for most of the individuals the I_{1548}/I_{1654} -ratio remains fairly constant over time (e.g., AE, AH, AM, AX and AY), while for others (e.g., BD) we observed a significant change over time – thus revealing a fair presentation of the results obtained.

To clarify why we choose to show this subset, we have added the latter explanation to the main text and extended the legend of the Figure 3 (**Revision 14**).

6. The most convincing evidence that the authors can provide would be to compare their results for “healthy” individuals with those of some who are not healthy, though this goal may be beyond the scope of this paper.

We fully agree with the Referee, and very much appreciate that he/she shares the same vision as we do. Unfortunately, as clearly pointed out by the Referee, a study assessing possible differences between non-symptomatic, healthy individuals and any kind of disease patients is beyond the scope of this work. This work aims to set the framework for any future applications of infrared fingerprinting to health monitoring. Indeed, we demonstrate that the within-person variability is lower than the between person variability over time – thus providing a foundation to possible applications in monitoring human health, as well as to detect possible disease conditions.

Fortunately, we can assure the Referee that we are already working on such disease detection case-control studies and will hopefully be able to share the results with the scientific community in the near future as well.

Minor Comments

1. Line 88: I think the word “variability” would be better than “instability”.

We changed the word according to the suggestion of the Referee.

2. Could a reference be included for the “Well-defined standardized protocols”

In our study, we did not follow any published operating procedures for blood collection and sample preparation as it did not fit our workflow. Therefore, we unfortunately cannot give a reference here. However, we added additional information to the Methods section, now explaining the procedures in more detail (**Revision 15**).

3. Figure 3 legend: The legend incorrectly identifies the I1548/I1654 ratio, but it appears to be the I1635/I1654 ratio.

We would like to thank the Referee for pointing out the mistake that we corrected in the revision (**Revision 16**).

4. Figure 4c: what do the different colors represent? What do the different symbols represent?

To allow for a clearer understanding of the Figure 4, we added an explanation to the figure caption providing a better description of the symbols used (**Revision 17**).

5. Figure 4d: The off-diagonal squares are so light that you cannot see them at first glance, especially in Figure 4d.

We would like to note to the Referee that the fact that the squares are hardly to see actually reflects the very high accuracy of the model. The colour reflects the density of predictions per class. The fact that the misclassification rate is so means that the colour of the off-diagonal elements is taken from the lower end of the colour map and therefore is very transparent.

6. Figure 4f,g: Are the axes correctly labeled? It appears to me that none of these individuals were correctly classified, since the axes do not match. So how can these be proper confusion matrices?

We would like to thank the Referee very much to point out this visualization error of ours. We have corrected the axis labelling (**Revision 18**).

7. Figure S2 has an X-axis from 1 to 7. Since subjects had up to 13 measures over 7 weeks, which subset of measures were selected for this figure? I realize it is the number of measure used to obtain the prediction for a person, but what subset of the 13 observations were used?

We used the same subset as was used for the results shown in Figure 4. The reasons why we chose this subgroup are the same, as explained in our answer above (please see answers to the point **5e**; see also **Revision 5**, and **13**).

Further, what are the values used for parameters for each classification method, even if they are the default values?

For the predictive modelling, we use advanced classification algorithms such as random forests and gradient boosting, which are associated with a big number of hyperparameters. In the caption of the Figure S3, we only report the value of the hyperparameters which was tuned (and thus different from the default) for optimizing the classification performance. For reproducibility, we also provide the version of the related Python packages. In this way, the

values of all hyperparameters can always be independently retrieved. To clarify this point even better, we additionally cite the web links to the description of the algorithms, including the full lists of hyperparameters used (**Revision 22**).

8. What do the colors mean in Table S1? What do the colors mean in Tables S3 and S4?

The colours in our tables do not have a deeper meaning. They are meant to be a guide to the eye to ease the overview of retrieved results. Large numbers correspond to green colours and low numbers to red colours. We added a short description to the respective Figure caption in the revised version of the Supplementary Information (**Revision 20**).

Additional changes (not prompted by Referee comments):

- All Figures and Tables were renumbered according to the newly introduced figures.
- We added the sentence “*However, new spectroscopic schemes allow to overcome current limitations in sensitivity and have the potential to significantly increase the range of detectable molecular concentrations.*”, along with two new references in the introduction - to reflect recent developments in the technology of infrared spectroscopy (**Revision 21**).
- We have corrected a mistake in Figure 2a. Instead of the inter-clinical variability of the serum, the corresponding data of plasma were shown. Now the correct data are shown.

REVIEWER COMMENTS

Reviewer #2 (Remarks to the Author):

I thank the reviewers for their corrections and for the inclusion of the power calculations.

A major issue lies with the water spectrum and as such nobody will be able to repeat or understand this experiment or indeed share in the information and data which I think is required by the journal without that spectrum or without buying the particular instrumentation from the supplier. Can the authors show how they will be able to comply with data sharing in light of this? If it is just a spectrum of water I can't understand what the issue is?

With reference to whether the spectrum is dry or drying the figure in S1 does not show this and this still hasn't been proven. If a sample is held in a cuvette for any period of time it will start drying especially at any liquid solid interface. Do the authors have actual spectral evidence of this as the figure in S1 only shows signal intensity which the authors show as evidence when what is needed is actual absorbance spectra to show that it hasn't dried - I imagine this is in the dataset already?

Reviewer #3 (Remarks to the Author):

The manuscript is much improved in content and stylistically. It now provides details that enable the reader to better understand what the authors did.

Minor Comments:

1. Results, Page 6: I suggest the following change, "This combined error was evaluated in a separate study by comparing blood samples obtained at 4 different clinics from 5 individuals within less than 4 hours.
2. The authors frequently refer to the "train and test set" for the classification procedures. Usually we say the training set and the test set.

Dear Referees,

We would like to thank you for your thoughts and comments that have helped us to improve our manuscript. In what follows, we address your comments in a point-by-point fashion. Generally, in response to your comments and suggestions, we have attained all the points raised and implemented corresponding changes that are tracked and enumerated (REV 1, REV 2, ...) in the revised manuscript.

In our response below, we would like to provide you with our answers to each point that you have raised.

Response to Reviewer #2

I thank the reviewers for their corrections and for the inclusion of the power calculations.

We would like to thank Reviewer 2 for appreciating the power calculations we included.

A major issue lies with the water spectrum and as such nobody will be able to repeat or understand this experiment or indeed share in the information and data which I think is required by the journal without that spectrum or without buying the particular instrumentation from the supplier.

We fully agree with Reviewer 2 that data sharing and reproducibility of measurements and experiments are crucial elements of the scientific work. For this reason, we provide all raw absorption spectra acquired with the MIRA-Analyzer as Supplementary Data. As discussed in more detail below, we believe that the raw absorption spectra are sufficient for data sharing purposes. To prove that the experiments can indeed be repeated without requiring this very particular spectrometer, we performed additional measurements with different instrumentation from another supplier. These data further confirm that qualitatively the same absorption spectra can be obtained with different FTIR measurement devices when the same sample is being measured. This comparison is now also included in the updated Supplementary Figure S1 (**Revision 1**).

To better understand our argumentation that the absorption spectra is sufficient for data sharing, it is insightful to evaluate how the absorption spectra are calculated and what kind of information it contains. A spectroscopic measurement is performed to obtain the optical properties (i.e. real and imaginary part of its refractive index) of a given sample. In FTIR spectroscopy usually only the absorbance A (i.e. imaginary part of the refractive index) of the sample is evaluated. This is obtained via $A = \log_{10}(I_R/I_S)$, where I_S denotes the spectral intensity of the sample measurement and I_R the spectral intensity of the reference measurement. Under ideal measurement conditions, the only difference between the sample and the reference measurement is that a sample was introduced. In this case, the intensity of the sample measurement would be $I_S = 10^{-A}I_R$. Consequently, the obtained absorbance is

independent of the characteristics of the measured power spectrum of the transmitted beam (given by the employed light source, optics, detectors, interferometer), and solely contains sample-related information, thus is independent of the employed spectrometer. Therefore, providing only the raw absorbance spectra (which we already uploaded and shared with the reviewers) is sufficient for data sharing and facilitates further data evaluation.

We admit that in experimental hands-on practice, a ‘perfect’ measurement does not exist, and thus every measurement will, to some extent, depend on the employed instrumentation. Yet, this is true for all kinds of measurements. The more practically relevant question is whether the influence of the choice of the measurement device is sufficiently small so that the obtained results can be compared between different devices. Given the advanced maturity and stability of FTIR instrumentation, this is indeed the case and we demonstrate this by performing **additional measurements**: We use the same quality control sera (same sample) and measure it with the automated MIRA-Analyzer (Micro-Biolytics), used throughout the measurements in this manuscript, and a Vertex 70 FTIR (Bruker) spectrometer. The updated version of Supplementary Information Figure 1 shows that both devices yield qualitatively the same raw absorption spectra, which demonstrates that the results obtained by different devices are indeed comparable (**Revision 1**). The reason for choosing the MIRA-Analyzer for the measurements presented in this manuscript lays in the higher reproducibility and quality of the spectra as compared to the Vertex 70 FTIR. This becomes evident in more detailed comparison between the two instruments that we prepared for review purposes (see Appendix of the document). For review purposes, we also provide the raw data that was used for this comparison (i.e. raw absorption spectra in the case of the measurements performed with the MIRA-Analyzer, as well as the power spectra of the reference/background and sample measurements together with the resulting absorption spectra in the case of the Vertex 70 FTIR – see ‘NCOMMS-20-02649C_additional data for review purposes.zip’.

Our results reveal that any researcher with access to an infrared spectrometer that can reliably acquire spectra of liquid samples in transmission (there are several manufactures of FTIR spectrometer and several suppliers of liquid measurement cuvettes on the market) should be able to obtain similar, thus comparable results. We believe to have designed and performed a study that would be of general relevance, as well as of general applicability, not specific to the spectrometer we chose to use here. The results of the additional measurements performed for this revision reflects and confirms this.

Can the authors show how they will be able to comply with data sharing in light of this?

We agree and support the sharing of experimental results with the scientific community. Towards that end, we also provided the data of the raw absorbance spectra of liquid serum and plasma. As described above, we consider this sufficient for the ability of any researcher to repeat these evaluations.

If it is just a spectrum of water i cant understand what the issue is?

Very unfortunately, we are not fully sure whether we fully understood the issue raised by the reviewer here. Assuming that she/he is referring to the water spectrum that we used for correcting the obtained, measured spectra, this would be our reply:

The water spectrum for correcting the obtained spectra of liquid serum and plasma was obtained from the literature (*Segelstein, D. J. The complex refractive index of water. (1981)*) and the whole correction procedure is described in detail in the Methods section and the SI. The literature reference was also added to the SI (**Revision 2**).

In addition to the above, we would like to point out that the water correction for the pre-processing of the infrared absorption spectra of liquid samples was not invented by us nor used for the very first time in our work. We followed the guidelines of the following peer-reviewed published article, that we also cite in our manuscript:

Yang, H., Yang, S., Kong, J., Dong, A. & Yu, S. Obtaining information about protein secondary structures in aqueous solution using Fourier transform IR spectroscopy. Nat. Protoc. 10, 382–396 (2015).

With reference to whether the spectrum is dry or drying the figure in S1 does not show this and this still hasn't been proven. If a sample is held in a cuvette for any period of time it will start drying especially at any liquid solid interface.

Unfortunately, we do not fully understand the criticism raised here. First of all, the intention of Fig. S1 is to show that the samples, while being measured, are still in the liquid phase. We do not understand why Reviewer 2 would still think that the sample may be dried when measured. A liquid measurement cuvette is a fully closed system (except during active filling and emptying of the cuvette) in the MIRA-Analyzer (Microbiolytics GmbH) setup, specially designed for analytical infrared fingerprint measurements of liquids (<https://www.microbiolytics.com/>). It would be impossible for water to evaporate out of the closed, tight fluidic system filled with either sample, transport media, or washing solutions. Thus, the whole tubing system of the automated sample injection unit – consisting not only of the measurement cuvette but also the tubing system for sample injection and system washing - stays filled with fluids during the measurements we perform (on the scale of minutes).

Although we acknowledge that it has previously been more common to perform measurements on dried serum or dried plasma, and that the FTIR community started with measurements of dry specimens, there are several publications in which infrared spectroscopy of liquid serum or plasma samples have already been performed and published in peer-reviewed journals.

Please find the examples of such studies listed below, as well as also cited as reference # 20, 40, and 49 in our main part of the manuscript.

- *Brandstetter, M., Volgger, L., Genner, A., Jungbauer, C. & Lendl, B. Direct determination of glucose, lactate and triglycerides in blood serum by a tunable quantum cascade laser-based mid-IR sensor. Appl. Phys. B 110, 233–239 (2013).*
- *Fabian, H., Lasch, P. & Naumann, D. Analysis of biofluids in aqueous environment based on mid-infrared spectroscopy. J. Biomed. Opt. 10, 031103 (2005).*
- *Lasch, P., Beekes, M., Fabian, H. & Naumann, D. Antemortem Identification of Transmissible Spongiform Encephalopathy (TSE) from Serum by Mid-infrared*

- Spectroscopy. Handbook of Vibrational Spectroscopy (2001).*
doi:10.1002/0470027320.s8925
- *Sala, A. et al. Rapid analysis of disease state in liquid human serum combining infrared spectroscopy and “digital drying”. J. Biophotonics (2020).*
doi:10.1002/jbio.202000118

If required, we can provide the Reviewer with more literature on this topic.

Do the authors have actual spectral evidence of this as the figure in S1 only shows signal intensity which the authors show as evidence when what is needed is actual absorbance spectra to show that it hasn't dried - I imagine this is in the dataset already?

We would like to explain that Fig. S1 was meant to explain why the spectral measurement of the cuvette filled with liquid is sufficient to be able to evaluate whether the sample is still in the liquid phase. The grey line in Fig. S1 shows the power spectra of the empty measurement cuvette, which corresponds to a completely ‘dry’ measurement compartment. As soon as an aqueous liquid (e.g. water or serum) is filled into the cuvette, the spectral intensity drops significantly at $\sim 1640\text{ cm}^{-1}$ and $\sim 3400\text{ cm}^{-1}$ (dark blue line).

This demonstrates that even a slight ‘drying’ of any sample in the cuvette would result in a significant change of signal intensity, and would therefore be readily detectable. Consequently, this also addresses Reviewer 2’s concern addressed above: In case the sample in the cuvette would be either dry or drying we would immediately be able to distinguish it from a fully liquid sample. This explanation can also be found in the figure caption of the revised Fig. S1.

To illustrate this further, we provide a measurement for which the measurement compartment was not fully liquid on purpose, which shows that this results in an extensive change of the absorbance spectrum (see Appendix).

Response to Referee #3

The manuscript is much improved in content and stylistically. It now provides details that enable the reader to better understand what the authors did.

We would like to thank the Referee for appreciating our efforts to revise the manuscript.

Minor Comments:

- 1. Results, Page 6: I suggest the following change, “This combined error was evaluated in a separate study by comparing blood samples obtained at 4 different clinics from 5 individuals within less than 4 hours.**
- 2. The authors frequently refer to the “train and test set” for the classification procedures. Usually we say the training set and the test set.**

We would like to thank the referee for these suggestions.

In the revised manuscript we have implemented these according to the suggestions of Reviewer 3 (**Revision 3 and Revision 4**).

REVIEWERS' COMMENTS

Reviewer #2 (Remarks to the Author):

The authors have shown experimental evidence in order to make this paper publishable and I would like to thank them for it

However they still need to change and check that all instances they refer to Raw data it is actually Raw. For example Figure 1 does not show raw data it shows data after analysis for water subtraction. This issue is replicated throughout the manuscript